# InTEn-LOAM: Intensity and Temporal Enhanced LiDAR Odometry and Mapping

Shuaixin Li [1,*,†], Bin Tian [2,†], Xiaozhou Zhu [1], Jianjun Gui [1], Wen Yao [1] and Guangyun Li [3]

1. The National Innovation Institute of Defense Technology, PLA Academy of Military Sciences, Beijing 100079, China
2. The State Key Laboratory of Management and Control for Complex Systems, Institute of Automation, Chinese Academy of Sciences, Beijing 100081, China
3. The Department of Geospatial Information, PLA Information Engineering University, Zhengzhou 450001, China
* Correspondence: lsx_navigation@sina.com
† These authors contributed equally to this work.

**Abstract:** Traditional LiDAR odometry (LO) systems mainly leverage geometric information obtained from the traversed surroundings to register lazer scans and estimate LiDAR ego-motion, while they may be unreliable in dynamic or degraded environments. This paper proposes InTEn-LOAM, a low-drift and robust LiDAR odometry and mapping method that fully exploits implicit information of lazer sweeps (i.e., geometric, intensity and temporal characteristics). The specific content of this work includes method innovation and experimental verification. With respect to method innovation, we propose the cylindrical-image-based feature extraction scheme, which makes use of the characteristic of uniform spatial distribution of lazer points to boost the adaptive extraction of various types of features, i.e., ground, beam, facade and reflector. We propose a novel intensity-based point registration algorithm and incorporate it into the LiDAR odometry, enabling the LO system to jointly estimate the LiDAR ego-motion using both geometric and intensity feature points. To eliminate the interference of dynamic objects, we propose a temporal-based dynamic object removal approach to filter them out in the resulting points map. Moreover, the local map is organized and downsampled using a temporal-related voxel grid filter to maintain the similarity between the current scan and the static local map. With respect to experimental verification, extensive tests are conducted on both simulated and real-world datasets. The results show that the proposed method achieves similar or better accuracy with respect to the state-of-the-art in normal driving scenarios and outperforms geometric-based LO in unstructured environments.

**Keywords:** SLAM; LiDAR odometry; dynamic removal; point intensity; scan registration

## 1. Introduction

Autonomous robots and self-driving vehicles must have the ability to localize themselves and intelligently perceive external surroundings. Simultaneous localization and mapping (SLAM) focuses on the issue of vehicle localization and navigation in unknown environments, which plays a major role in many autonomous driving and robotics-related applications, such as mobile mapping [1], space exploration [2], robot localization [3] and high-definition map production [4]. In accordance with the on-board perceptional sensors, it can be roughly classified into two categories, i.e., camera-based and LiDAR (Light detection and ranging)-based SLAM. Compared with images, LiDAR point clouds are invariant with respect to the changing illumination and are sufficiently dense for 3D reconstruction tasks. Accordingly, LiDAR SLAM solutions have become a preferred choice for self-driving car manufacturers compared to vision-based solutions [5–7]. Note that a complete *SLAM solution* usually composes the front-end and back-end, which are in charge of ego-estimation by tracking landmarks and global optimization by recognizing loop-closures, respectively.

The front-end part is also referred to as *odometry solution*, though both *SLAM* and *odometry* have self-localization abilities in unknown scenes and mapping abilites with respect to traversprincipled environments. For instance, though both LOAM [8] and LeGO-LOAM [9] enable one to perform low-drift and real-time ego-estimation and mapping, only LeGO-LOAM can be referred to as a complete SLAM solution since it is a loop closure-enabled system.

Remarkable progress has occurred in LiDAR-based SLAM over recent decade [8–13]. State-of-the-art solutions have shown remarkable performances, especially in structured urban and indoor scenes. Recent years have seen solutions for more intractable problems, e.g., fusion with multiple sensors [14–18], adapting to cutting-edge solid-state LiDAR [19], global localization [20], improving the efficiency of optimization back-end [21,22], etc., yet many issues remain unsolved. Specifically, most conventional LO solutions currently ignore intensity information from the reflectance channel, though they reveal reflectivities of different objects in the real world. An efficient incorporation approach making use of point intensity information is still an open problem since the intensity value is not as straightforward as the range value. It is a value with many factors, including the material of target surface, the scanning distance, the lazer incidence angle, as well as the transmitted energy. In addition, the lazer sweep represents a snapshot of surroundings and thus moving objects, such as pedestrians, vehicles, etc., may be scanned. These dynamic objects result in 'ghosting points' in the accumulated points map and increase the probability of incorrect matching during scan registration, which may deteriorate the localization accuracy of LO. Moreover, improving the robustness of point registration in some geometric-degraded environments, e.g., long straight tunnel, is also a topic worthy of in-depth discussion. In this paper, we present InTEn-LOAM (as shown in Figure 1) to cope with the aforementioned challenges. The main contributions of our work are summarized four-fold as follows:

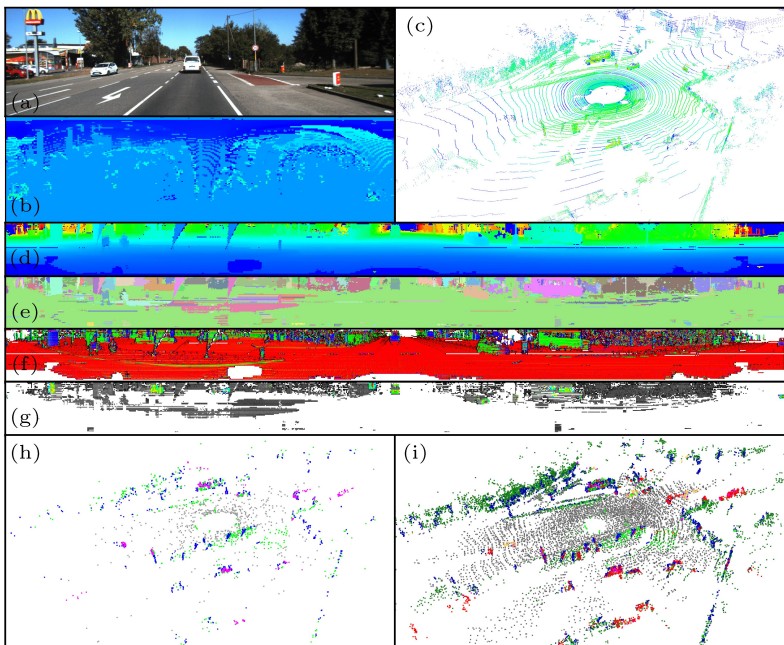

**Figure 1.** Overview of the proposed InTEn-LOAM system. (**a**) The color image from the on-board camera. (**b**) The projected scan-context segment image. (**c**) The raw point cloud from the on-board Velodyne HDL-64 LiDAR colored according to intensity. (**d**) The projected cylindrical range image colored according to depth. (**e**) The segmented label image. (**f**) The estimated normal image (x, y, z). (**g**) The intensity image of non-ground points. Only reflector features are colored. (**h**) Various types of feature (ground, facade, beam, reflector) extracted from the current lazer scan. (**i**) The current point features align with the local feature map that is in use so far (dynamic object in the current scan).

- We propose an efficient range-image-based feature extraction method that is able to adaptively extract features from the raw lazer scan and categorize them into four different types in real time.
- We propose a coarse-to-fine, model-free method for online dynamic object removal enabling the LO system to build a purely static map by removing all dynamic outliers in raw scans.
- We propose a novel intensity-based points registration algorithm that directly leverages reflectance measurements to align point clouds, and we introduce it into the LO framework to achieve jointly pose estimation utilizing both geometric and intensity information.
- Extensive experiments are conducted to evaluate the proposed system. Results show that InTEn-LOAM achieves similar or better accuracy in comparison with state-of-the-art LO systems and outperforms them in unstructured scenes with sparse geometric features.

## 2. Related Work

### 2.1. Point Cloud Registration and LiDAR Odometry

Point cloud registration is the most critical problem in LiDAR-based autonomous driving, which is centered on finding the best relative transformation of point clouds. Existing registration techniques can be either categorized into feature-based and scan-based methods [23] in terms of the type of data or local and global methods [24] in terms of the registration reference. Though the local registration requires a good initial transformation, it has been widely used in LO solutions since sequential LiDAR sweeps commonly share large overlap and a coarse initial transformation can be readily predicted.

For feature-based approaches, different types of encoded features, e.g., FPFH (fast point feature histogram) [25], CGF (compact geometric feature) [26] and arbitrary shapes are extracted to establish valid data associations. LOAM [8] is one of the pioneering works of feature-based LO, which extracts plane and edge features based on the sorted smoothness of each point. Many follow-up works follow the proposed feature extraction scheme [16–19]. For example, LeGO-LOAM [9] additionally segmented ground to bound the drift in the ground norm direction. MULLS (multi-metric linear least square) [27] explicitly classifies features into six types, (facade, ground, roof, beam, pillar and encoded points) using the principal component analysis (PCA) algorithm and employs the least square algorithm to estimate the ego-motion, which remarkably improves the LO performance, especially in unstructured environments. Yin, et al. [28] propose a convolutional auto-encoder (CAE) to encode feature points to conduct a more robust point association.

Scan-based local registration methods iteratively assign correspondences based on the closest-distance criterion. The iterative closest point (ICP) algorithm, introduced by [29], is the most popular scan registration method. Many variants of ICP have been derived over the past three decades, such as Generalized ICP (GICP) [30] and improved GICP [31]. Many LO solutions apply variants of ICP to align scans for their simplicity and low computational complexity. For example, Moosmann, et al. [32] employ standard ICP, while Palieri, et al. [15] and Behley, et al. [10] employ GICP and normal ICP. The normal distributions transform (NDT) method, first introduced by [33], is another popular scan-based approach, in which surface likelihoods of the reference scan are used for scan matching. Because of that, there is no need for computationally expensive nearest-neighbor searching in NDT, making it more suitable for LO with large-scale map points [12–14].

### 2.2. Fusion with Point Intensity

Some works have attempted to introduce the intensity channel into scan registration. Inspired by GICP, Servos [34] proposes the multichannel GICP (MCGICP), which integrates color and intensity information into the GICP framework by incorporating additional channel measurements into the covariances of points. In [35], a data-driven intensity calibration approach is presented to acquire a pose-invariant measure of surface reflectivity.

Based on that, Wang [36] establishes voxel-based intensity constraints to complement the geometric-only constraints in the mapping thread of LOAM. Pan [27] assigns higher weights for associations with similar intensities to suppress the effect of outliers adaptively. In addition, the end-to-end learning-based registration framework, named Deep VCP (virtual corresponding points) [37], is proposed, achieving accuracy comparable to the prior state-of-the-art. The intensity channel is used to find stable and robust feature associations, which are helpful with respect to avoiding the interference of negative true matchings.

### 2.3. Dynamic Object Removal

A good amount of learning-based works related to dynamic removal have been reported in [38]. In general, the trained model is used to predict the probability score that a point originated from dynamic objects. The model-based approaches enable one to filter out the dynamics independently, but they also require laborious training tasks and the segmentation performance is highly dependent on the training dataset.

Traditional model-free approaches rely on differences between the current lazer scan and previous scans [39–41]. Though they are convenient and straightforward, only points that have fully moved outside their original position can be detected/removed.

## 3. Materials and Methods

The proposed framework of InTEn-LOAM consists of 5 submodules, i.e., feature extraction filter (FEF), scan-to-scan registration (S2S), scan-to-map registration (S2M), temporal-based voxel filter (TVF) and dynamic object removal (DOR) (see Figure 2). Following LOAM, LiDAR odometry and mapping are executed on two parallel threads to improve the running efficiency.

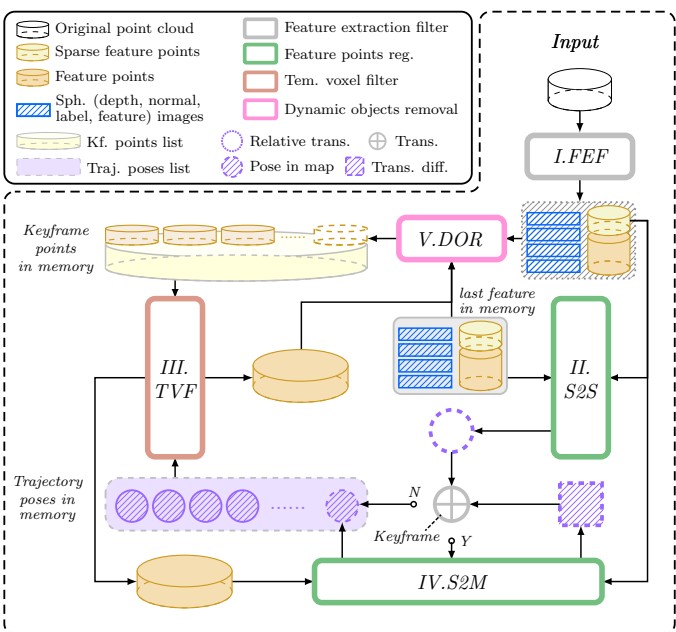

**Figure 2.** Overall workflow of InTEn-LOAM.

### 3.1. Feature Extraction Filter

The workflow of FEF is summarized in Figure 3, which corresponds to the gray block in Figure 2. The FEF receives a raw scan frame and outputs four types of features, i.e., ground, facade, beam and reflector and two types of cylindrical images, i.e., range and label images.

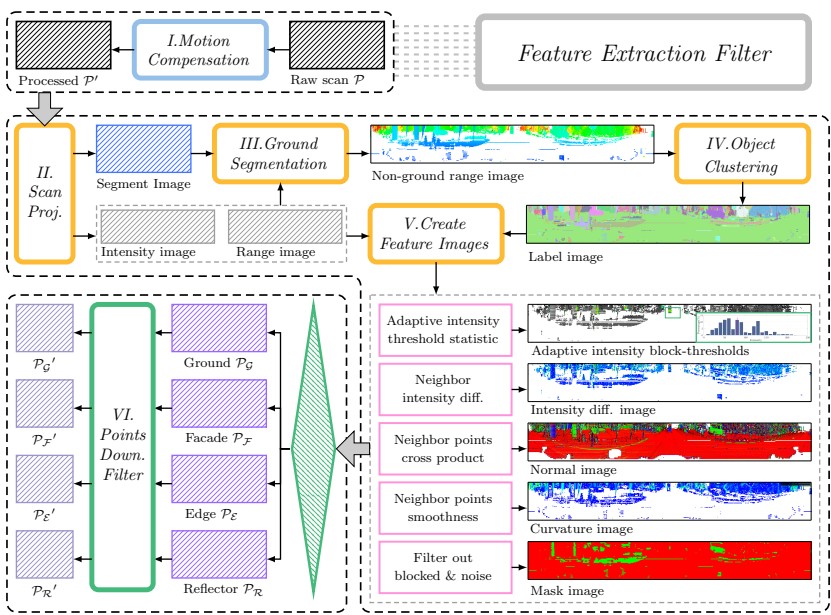

**Figure 3.** The workflow of FEF.

### 3.1.1. Motion Compensation

Given the point-wise timestamp of a scan $\mathcal{P}$, the reference pose for a point $\mathbf{p}_i \in \mathcal{P}$ at timestamp $\tau_i$ can be interpolated by the relative transformation $\mathbf{T}_{e,s} = [\mathbf{R}_{e,s}, \mathbf{t}_{e,s}]$ under the assumption of uniform motion:

$$\mathbf{T}_{s,i} = [\text{slerp}(\mathbf{R}_{e,s}, s_i)^\top, -s_i \cdot \mathbf{T}_{e,s}^{-1} \cdot \mathbf{t}_{e,s}], \tag{1}$$

where $\text{slerp}(\cdot)$ represents the spherical linear interpolation. The time ratio $s_i$ is $s_i = \frac{\tau_i - \tau_s}{\tau_e - \tau_s}$, where $\tau_s$, $\tau_e$ stand for the start and end timestamps of the lazer sweep, respectively. Then, the distorted points can be deskewed by transforming to the start timestamp $\mathbf{T}_{s,i} \cdot \mathbf{p}_i \in \mathcal{P}'$. Note that the current relative transformation $\mathbf{T}_{e,s}$ can be estimated by registering lazer scans in timestamps $\tau_e$ and $\tau_s$, which will be described in Section 3.4.

### 3.1.2. Scan Preprocess

The undistorted points $\mathcal{P}'$ are first preprocessed. The main steps are as below:

*I. Scan projection.* Since raw 3D lazer points are disordered, it is time-consuming to search one specific point and its neighbors. Scan projection is a good way to manage 3D points with an ordered 2D matrix, which facilitates searching a local region of interest points. $\mathcal{P}'$ is projected into a cylindrical plane to generate range and intensity images, i.e., $\mathcal{D}$ and $\mathcal{I}$ (see Figure 1d,e). A point with 3D coordinates in the lazer frame $\mathbf{p}_i = [x, y, z]^\top$ can be projected as a cylindrical image pixel $[u, v]^\top$ by:

$$\begin{pmatrix} u \\ v \end{pmatrix} = \begin{pmatrix} [1 - \arctan(y, x) \cdot \pi^{-1}] \cdot \frac{w}{2} \\ (\arcsin(\frac{z}{\sqrt{x^2+y^2+z^2}}) + \theta_d) \cdot \frac{h}{\theta} \end{pmatrix}, \tag{2}$$

where $\theta = \theta_d + \theta_t$ is the vertical field-of-view of the LiDAR (vertical angle of the bottom lazer ray $\theta_d$ plus that of the top lazer ray $\theta_t$), and $w, h$ are the width and height of the resulting image. In $\mathcal{D}$ and $\mathcal{I}$, each pixel contains the smallest range and the largest reflectance of scanning points falling into the pixel, respectively. In addition, $\mathcal{P}'$ is also preprocessed as a segment image $\mathcal{S}$ (see Figure 1b) according to azimuthal and radial directions of 3D points, and each pixel contains the lowest $z$. The former converter is the same as $u$ in Equation (2), while the latter is equally spaced with the distance interval $\Delta\rho$:

$$\rho = \lfloor \sqrt{(x^2 + y^2 + z^2)}/\Delta\rho \rfloor, \tag{3}$$

where $\lfloor \cdot \rfloor$ indicates a rounding down operator. Note that the size of $\mathcal{S}$ is not the same as $\mathcal{D}$.

*II. Ground segmentation.* The method from [42] is applied in this paper with the input of segment image $S$. Each column of $\mathcal{S}$ is fitted as a ground line $\mathbf{l}_i = a_i \cdot \rho + b_i$. Then, residuals can be calculated representing the differences between the predicted and the observed $z$:

$$r(u,v) = \mathbf{l}_i(\mathcal{D}(u,v)) - \mathcal{D}(u,v) \cdot \sin(\theta_v), \tag{4}$$

where $\theta_v$ indicates the vertical angle of the $v$th row in $\mathcal{D}$. Pixels with residuals smaller than the threshold $\text{Th}_g$ will be marked as ground pixels with label identity 1. Then, the non-ground image $\mathcal{D}'$ can be generated.

*III. Object clustering.* After the ground segmentation, the angle-based object clustering approach from [43] is used in $\mathcal{D}'$ to group non-ground pixels into different clusters with identified labels and generate a label image $\mathcal{L}$ (see the label image in Figure 3).

*IV. Create feature images.* We partition the intensity image $\mathcal{I}$ into $M \times N$ blocks and establish intensity histograms for each block. The extraction threshold $\text{Th}_{I,n}$ of each intensity block is adaptively determined by taking the median of the histogram. In addition, the intensity difference image $\mathcal{I}_\Delta$, the normal image $\mathcal{N}$ and the curvature image $\mathcal{C}$ are created by:

$$
\begin{aligned}
\mathcal{I}_\Delta(u,v) &= \mathcal{I}(u,v) - \mathcal{I}(u,v+1), \\
\mathcal{N}(u,v) &= (\Pi[\mathcal{D}(u+1,v)] - \Pi[\mathcal{D}(u,v)]) \\
&\quad \times (\Pi[\mathcal{D}(u,v+1)] - \Pi[\mathcal{D}(u,v)]), \\
\mathcal{C}(u,v) &= \\
&\frac{1}{N \cdot \mathcal{D}(u,v)} \cdot \sum_{i,j \in N} (\mathcal{D}(u,v) - \mathcal{D}(u+i,v+j))
\end{aligned}
\tag{5}
$$

where $\Pi[\cdot] : \mathcal{D} \mapsto \mathcal{P}$ denotes the mapping function from a range image pixel to a 3D point. $N$ is the neighboring pixel count. Furthermore, pixels in the cluster with fewer than 15 points are marked as noises and blocked. All the valid-or-not flag is stored in a binary mask image $\mathcal{B}$.

### 3.1.3. Feature Extraction

According to the above feature images, pixels of four categories of features can be extracted. Then, 3D feature points, i.e., ground $\mathcal{P}_\mathcal{G}$, facade $\mathcal{P}_\mathcal{F}$, edge $\mathcal{P}_\mathcal{E}$ and reflector $\mathcal{P}_\mathcal{R}$, can be obtained per the pixel-to-point mapping relationship. Specifically,

- Points correspond to pixels that meet $\mathcal{L}(u,v) = 1$ and $\mathcal{B}(u,v) \neq 0$ are categorized as $\mathcal{P}_\mathcal{G}$.
- Points correspond to pixels that meet $\mathcal{I}.(u,v) > Th_{\Delta I}$ and $\mathcal{B}(u,v) \neq 0$ are categorized as $\mathcal{P}_\mathcal{R}$. In addition, points in pixels that meet $\mathcal{I}(u,v) > Th_{I,n}$ and their neighbors are all included in $\mathcal{P}_\mathcal{R}$ to keep the local intensity gradient of reflector features.
- Points correspond to pixels that meet $\mathcal{C}(u,v) > Th_E$ and $\mathcal{B}(u,v) \neq 0$ are categorized as $\mathcal{P}_\mathcal{E}$.
- Points correspond to pixels that meet $\mathcal{C}(u,v) < Th_F$ and $\mathcal{B}(u,v) \neq 0$ are categorized as $\mathcal{P}_\mathcal{F}$.

To improve the efficiency of scan registration, the random downsample filter (RDF) is applied on $\mathcal{P}_\mathcal{G}$ and $\mathcal{P}_\mathcal{R}$ to obtain $N_\mathcal{G}$ downsampled edge features $\mathcal{P}_\mathcal{G}'$ and $N_\mathcal{R}$ facade features $\mathcal{P}_\mathcal{R}'$. To obtain $N_\mathcal{E}$ refined edge features $\mathcal{P}_\mathcal{E}'$ and $N_\mathcal{F}$ refined facade features $\mathcal{P}_\mathcal{F}'$, the non-maximum suppression (NMS) filter based on point curvatures is applied on $\mathcal{P}_\mathcal{E}$ and $\mathcal{P}_\mathcal{F}$. Note that the thresholds above are empirical and all of them are consistently set as default for common scenarios.

### 3.2. Intensity-Based Scan Registration

Similar to the geometric-based scan registration, given the initial guess of the transformation $\bar{\mathbf{T}}_{t,s}$ from source points $\mathcal{P}_s$ to target points $\mathcal{P}_t$, we try to estimate the LiDAR motion

$\mathbf{T}_{t,s}$ by matching the local intensities of the source and target. In the case of geometric feature registration, the motion estimation is solved through nonlinear iterations by minimizing the sum of Euclidean distances from each source feature to their correspondence in the target scan. In the case of reflecting feature registration, however, we minimize the sum of intensity differences instead. The fundamental idea of the intensity-based point cloud alignment method proposed in this paper is to make use of the similarity of intensity gradients within the local region of lazer scans to achieve scan matching.

Because of the discreteness of the lazer scan, sparse 3D points in a local area are not continuous, causing the intensity values of the lazer sweep to be non-differentiable. To solve this issue, we introduce a continuous intensity surface model using the local support characteristic of the B-spline basis function. A simple intensity surface example is shown in Figure 4.

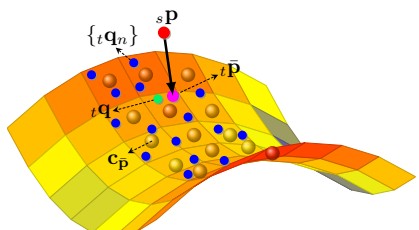

**Figure 4.** A simple example of B-spline intensity surface model. The grid surface depicts the modeled continuous intensity surface with colors representing intensities and spheres in the center of surface grids representing control points of the B-spline surface model. $_s\mathbf{p}$ denotes the selected point and $\{_t\mathbf{q}_n\}$ denotes query points. $_s\mathbf{p}$ is transformed to the reference frame of $\{_t\mathbf{q}_n\}$ and denoted as $_t\bar{\mathbf{p}}$. $_t\mathbf{q}$ denotes the nearest neighboring query point of $_t\bar{\mathbf{p}}$.

### 3.2.1. B-Spline Intensity Surface Model

The intensity surface model presented in this paper uses the uniformly distributed knots of the B-spline; thus, the B-spline is defined fully by its degree [44]. Specifically, the intensity surface is a space spanned by three $d$-degree B-spline functions on the orthogonal axes and each B-spline is controlled by $d+1$ knots on the axis. Mathematically, the B-spline intensity surface in local space is a scalar-valued function $^-(\mathbf{p}) : \mathbb{R}^3 \to \mathbb{R}$, which builds the mapping relationship between a 3D point $\mathbf{p} = [x, y, z]^\top$ and its intensity value. The mapping function is defined by the tensor product of three B-spline functions and control points $c_{i,j,k} \in C$ in the local space:

$$
\begin{aligned}
^-(\mathbf{p}) &= \sum_{i=0}^{d+1} \sum_{j=0}^{d+1} \sum_{k=0}^{d+1} c_{i,j,k} b_i^d(x) b_j^d(y) b_k^d(z) \\
&= \mathrm{vec}(\mathbf{b}_x^d \otimes \mathbf{b}_x^d \otimes \mathbf{b}_z^d)^\top \cdot \mathrm{vec}(C) \\
&= \boldsymbol{\phi}(\mathbf{p})^\top \cdot \mathbf{c}
\end{aligned}
\tag{6}
$$

where $\mathbf{b}^d$ is the $d$ degree B-spline function. We use the vectorization operator $\mathrm{vec}(\cdot)$ and Kronecker product operator $\otimes$ to transform the above equation in the form of matrix multiplication. In this paper, the cubic ($d = 3$) B-spline function is employed.

### 3.2.2. Observation Constraint

The intensity observation constraint is defined as the residual between the intensities of source points and their predicted intensities in the local intensity surface model. Figure 4 demonstrates how to predict the intensity on the surface patch for a reflector feature point. The selected point $_s\mathbf{p} \in \mathcal{P}_s$ with intensity measurement $\eta$ is transformed to the model frame by $_t\bar{\mathbf{q}} = \bar{\mathbf{T}}_{t,s} \cdot_s \mathbf{p}$. Then, the nearest point $_t\mathbf{q} \in \mathcal{P}_t$ and its R-neighbor points $_t\mathbf{q}_n \in \mathcal{P}_t, n = 1 \cdots N$ can be searched. Given the uniform space of the B-spline function $\kappa$, the neighborhood points $_t\mathbf{q}_n$ can be voxelized with the center $_t\mathbf{q}$ and the resolution

$\kappa \times \kappa \times \kappa$ to generate control knots $\mathbf{c}_{\bar{\mathbf{p}}}$ for the local intensity surface. The control knot takes the value of the average intensities of all points in a voxel. To sum up, the residual is defined as:

$$r_{\mathcal{I}}(\tilde{\mathbf{T}}_{t,s}) = [\boldsymbol{\phi}(\bar{\mathbf{T}}_{t,s} \cdot_s \mathbf{p})^{\top} \cdot \mathbf{c}_{\bar{\mathbf{q}}} - \eta]. \tag{7}$$

Stacking normalized residuals to obtain residual vector $\mathbf{r}_{\mathcal{I}}(\tilde{\mathbf{T}}_{t,s})$ and computing the Jacobian matrix of $\mathbf{r}_{\mathcal{I}}$ with respect to $\mathbf{T}_{t,s}$, denoted as $\mathbf{J}_{\mathcal{I}} = \partial \mathbf{r}_{\mathcal{I}} / \partial \mathbf{T}_{t,s}$. The constructed nonlinear optimization problem can be solved by minimizing $\mathbf{r}_{\mathcal{I}}$ toward zero using the L-M algorithm. Note that *Lie group* and *Lie algebra* are implemented for the 6-DoF transformation in this paper.

### 3.3. Dynamic Object Removal

The workflow of the proposed DOR is shown in Figure 5, which corresponds to the pink block in Figure 2. Inputs of the DOR filter include the current lazer points $\mathcal{P}_k$, the previous static lazer points $\mathcal{P}_{s,k-1}$, the local map points $\mathcal{M}_k$, the current range image $\mathcal{D}_{\mathcal{P}_k}$, the current label image $\mathcal{L}_{\mathcal{P}_k}$ and the estimated LiDAR pose in the world frame $\tilde{\mathbf{T}}_{w,k}$. The filter divides $\mathcal{P}_k$ into two categories, i.e., the dynamic $\mathcal{P}_{d,k}$ and static $\mathcal{P}_{s,k}$. Only static points will be appended into the local map for map update. The DOR filter introduced in this paper exploits the similarity of point clouds in the adjacent time domain for dynamic point filtering and verifies dynamic objects based on the segmented label image.

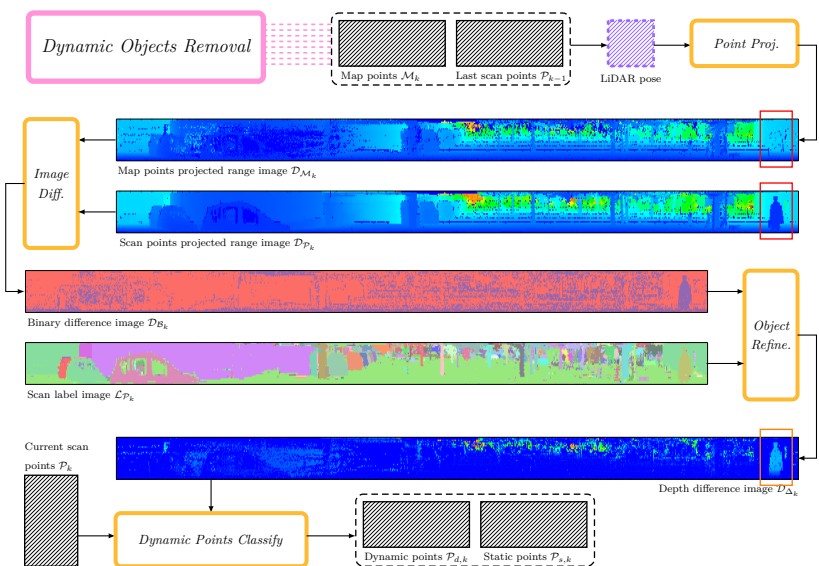

**Figure 5.** The workflow of DOR.

#### 3.3.1. Rendering Range Image for the Local Map

Both downsampling with coarse resolution and uneven distribution of map points may result in pixel holes in the rendered range image. Considering the great similarity of successive lazer sweeps in the time domain, we use both the local map points $\mathcal{M}_k$ and the previous static lazer points $\mathcal{P}_{s,k-1}$ to generate the to-be-rendered map points $\mathcal{E}_k$:

$$\mathcal{E}_k = \mathbf{T}_{w,k}^{-1} \cdot \mathcal{M}_k \cup \mathbf{T}_{k,k-1} \cdot \mathcal{P}_{s,k-1}. \tag{8}$$

The rendered image $\mathcal{D}_{\mathcal{M}_k}$ and the current scan image $\mathcal{D}_{\mathcal{P}_k}$ are shown in the second and third rows of Figure 5. A pedestrian (see red rectangle in Figure 5) can be clearly distinguished in $\mathcal{D}_{\mathcal{P}_k}$ but not in $\mathcal{D}_{\mathcal{M}_k}$.

### 3.3.2. Temporal-Based Dynamic Points Searching

Dynamic pixels in $\mathcal{D}_{\mathcal{P}_k}$ can be coarsely screened out in accordance with the depth differences between $\mathcal{D}_{\mathcal{P}_k}$ and $\mathcal{D}_{\mathcal{M}_k}$. In particular, if the depth difference at $[u,v]^\top$ is larger than the threshold $Th_{\Delta d}$, the pixel will be marked as dynamic. Consecutively, we can also generate a binary image $\mathcal{D}_{\mathcal{B}_k}$ indicating whether the pixel is dynamic or not:

$$
\begin{aligned}
\mathcal{D}_{\Delta_k}(u,v) = |\mathcal{D}_{\mathcal{M}_k}(u,v) - \mathcal{D}_{\mathcal{P}_k}(u,v)| > Th_{\Delta d}? \\
\mathcal{D}_{\mathcal{B}_k}(u,v) = 1 : 0,
\end{aligned}
\tag{9}
$$

where $\mathcal{D}_{\mathcal{M}_k}(u,v) \neq 0$ and $\mathcal{D}_{\mathcal{P}_k}(u,v) \neq 0$. An example of $\mathcal{D}_{\mathcal{B}_k}$ is shown in the fourth row of Figure 5, in which red pixels represent the static and purple pixels represent the dynamic. To improve the robustness of the DOR filter to different point depths, we use the adaptive threshold $Th_{\Delta d} = s_d \cdot \mathcal{D}_{\mathcal{P}_k}(u,v)$, where $s_d$ is a constant coefficient.

### 3.3.3. Dynamic Object Validation

It can be seen from $\mathcal{D}_{\mathcal{B}_k}$ that it generates numerous false positive (FP) dynamic pixels using pixel-by-pixel depth comparison. To handle the above issue, we utilize the label image to validate the dynamic according to the fact that points originating from the same object should have the same status label. We denote the pixel number of a segmented object and the dynamic pixel number as $N_i$ and $N_{d,i}$, which can be counted from $\mathcal{L}_{\mathcal{P}_k}$ and $\mathcal{D}_{\mathcal{B}_k}$, respectively. Two basic assumptions generally hold in terms of dynamic points, i.e., (I) ground points cannot be dynamic; (II) the percentage of FP dynamic pixels in a given object will not be significant. According to the above assumptions, we can validate dynamic pixels at the object level:

$$
\begin{aligned}
\frac{N_{d,i}}{N_i} \geq Th_N \ \& \ \mathcal{L}_{\mathcal{P}_k}(u,v) \neq 1? \\
\mathcal{D}_{\Delta_k}(u,v) = \mathcal{D}_{\Delta_k}(u,v) : 0.
\end{aligned}
\tag{10}
$$

Equation (10) indicates that only an object that is marked as a non-ground object or have a dynamic pixel ratio larger than the threshold will be recognized as dynamic. In $\mathcal{D}_{\Delta_k}$, pixels belonging to dynamic objects will retain the depth differences, while the others will be reset as 0. As can be observed in the depth difference image shown in the sixth row of Figure 5, though many FP dynamic pixels are filtered out after the validation, the true positive (TP) dynamic pixels from the moving pedestrian on the right side are still remarkable. Then, the binary image $\mathcal{D}_{\mathcal{B}_k}$ is updated by substituting the refined $\mathcal{D}_{\Delta_k}$ into Equation (9).

### 3.3.4. Points Classification

According to $\mathcal{D}_{\mathcal{B}_k}$, dynamic 3D points in extracted features can be marked using the mapping function $\Pi[\cdot] : \mathcal{D} \mapsto \mathcal{P}$. Since the static feature set is the complement of the dynamic feature set with respect to the full set of extracted features, the static features can be filtered by $\mathcal{P}_{s,k} = \mathcal{P}_k - \mathcal{P}_{d,k}$.

### 3.4. LiDAR Odometry

Given the initial guess $\bar{\mathbf{T}}_{k,k-1}$, extracted features, i.e., downsampled ground and reflector features $\mathcal{P}_{\mathcal{G}}'$ and $\mathcal{P}_{\mathcal{R}}'$, as well as refined edge and facade features $\mathcal{P}_{\mathcal{E}}'$ and $\mathcal{P}_{\mathcal{F}}'$, are utilized to estimate the optimal estimation of $\mathbf{T}_{k,k-1}$ and then the LiDAR pose $\mathbf{T}_{w,k}$ in the global frame is reckoned. The odometry thread corresponds to the green S2S block in Figure 2 and the pseudocode is shown in Algorithm 1. To improve the performance of geometric-only scan registration, the proposed LO incorporates reflector features and estimates relative motion by jointly solving multi-metric nonlinear optimization (NLO).

---

**Algorithm 1:** LiDAR Odometry

---

    **Input:** Extracted feature points $\mathcal{P}'_{\mathcal{G},k}$, $\mathcal{P}'_{\mathcal{F},k}$, $\mathcal{P}'_{\mathcal{E},k}$, $\mathcal{P}'_{\mathcal{R},k}$, initial transform $\bar{\mathbf{T}}_{k,k-1}$

    **Output:** estimated transform $\mathbf{T}_{k,k-1}$, reckoned pose $\mathbf{T}_{w',k}$

**1**   *reference feature points* $\mathcal{P}'_{\mathcal{G},k-1}$, $\mathcal{P}'_{\mathcal{F},k-1}$, $\mathcal{P}'_{\mathcal{E},k-1}$, $\mathcal{P}'_{\mathcal{R},k-1}$ *and the last LO reckoned pose* $\mathbf{T}_{w',k-1}$
    *can be loaded from the buffer;*

    `// Main`

**2**   **if** *the system is not initialized* **then**

**3**        $\mathbf{T}_{k,k-1} \leftarrow \mathbf{I}_{4\times4}$;

**4**        $\mathbf{T}_{w',k} \leftarrow \mathbf{I}_{4\times4}$;

**5**   **end**

**6**   **else**

**7**        **for** *a number of iterations* **do**

            `// Find feature associations by parallel threads`

**8**              $\mathbf{r}_{\mathcal{G}}, \mathbf{J}_{\mathcal{G}} \leftarrow$ `GroundAssoc(`$\bar{\mathbf{T}}_{k,k-1}, \mathcal{P}'_{\mathcal{G},k-1}, \mathcal{P}'_{\mathcal{G},k}$`)`;

**9**              $\mathbf{r}_{\mathcal{F}}, \mathbf{J}_{\mathcal{F}} \leftarrow$ `FacadeAssoc(`$\bar{\mathbf{T}}_{k,k-1}, \mathcal{P}'_{\mathcal{F},k-1}, \mathcal{P}'_{\mathcal{F},k}$`)`;

**10**            $\mathbf{r}_{\mathcal{E}}, \mathbf{J}_{\mathcal{E}} \leftarrow$ `EdgeAssoc(`$\bar{\mathbf{T}}_{k,k-1}, \mathcal{P}'_{\mathcal{E},k-1}, \mathcal{P}'_{\mathcal{E},k}$`)`;

**11**           $\mathbf{r}_{\mathcal{R}}, \mathbf{J}_{\mathcal{R}} \leftarrow$ `ReflectAssoc(`$\bar{\mathbf{T}}_{k,k-1}, \mathcal{P}'_{\mathcal{R},k-1}, \mathcal{P}'_{\mathcal{R},k}$`)`;

            `// Update relative transform by the nonlinear optimization`

**12**           $\tilde{\mathbf{T}}_{k-1,k} \leftarrow$ `MultiMetricNLO(J,r)`;

            `// Convergency`

**13**           *convergency* $\leftarrow$ `ConvergCond(`$\tilde{\mathbf{T}}_{k-1,k} \cdot \bar{\mathbf{T}}_{k,k-1}^{-1}$`)`;

            `// Update parameters`

**14**           $\mathbf{T}_{k,k-1} \leftarrow \tilde{\mathbf{T}}_{k-1,k}$;

**15**           $\mathbf{T}_{w',k} \leftarrow \mathbf{T}_{w',k-1} \cdot \mathbf{T}_{k,k-1}^{-1}$;

**16**           **if** *convergency* **then**

**17**              **break;**

**18**           **end**

**19**        **end**

**20**   **end**

---

### 3.4.1. Constraint Model

As shown in Figure 6, constraints are modeled as the point-to-model intensity difference (for reflector feature) and the point-to-line (for edge feature)/point-to-plane (for ground and facade feature) distance, respectively.

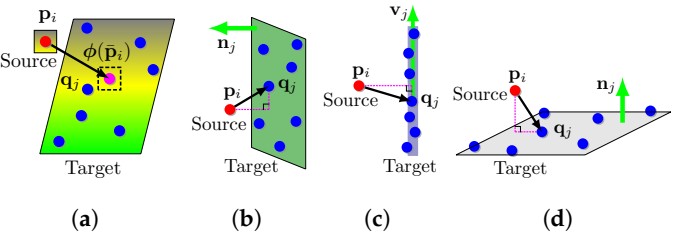

                  **(a)**            **(b)**          **(c)**            **(d)**

**Figure 6.** Overview of four different types of feature associations. (**a**) Reflector; (**b**) Facade; (**c**) Edge; (**d**) Ground feature association.

*I. Point-to-line constraint.* Let $\mathbf{p}_i \in \mathcal{P}'_{\mathcal{E},k}, i = 1 \cdots N_E$ be an edge feature point. The association of $\mathbf{p}_i$ is the line connected by $\mathbf{q}_j, \mathbf{q}_m \in \mathcal{P}'_{\mathcal{E},k-1}$, which represents the closest point of $\bar{\mathbf{T}}_{k-1,k} \cdot \mathbf{p}_i$ in $\mathcal{P}'_{\mathcal{E},k-1}$ and the closest neighbor in the preceding and following scan lines to the $\mathbf{q}_j$, respectively. The constraint equation is formulated as the point-to-line distance:

$$r_{\mathcal{E},i} = \|\mathbf{v}_j \times (\mathbf{T}_{k-1,k} \cdot \mathbf{p}_i)\|,$$
$$\mathbf{v}_j = \frac{\mathbf{q}_j - \mathbf{q}_m}{\|\mathbf{q}_j - \mathbf{q}_m\|}. \tag{11}$$

The $N_E \times 1$ edge feature error vector $\mathbf{r}_{\mathcal{E}}$ is constructed by stacking all the normalized edge residuals (Line 10).

*II. Point-to-plane constraint.* Let $\mathbf{p}_i \in \mathcal{P}'_{\mathcal{F},k}, (\mathcal{P}'_{\mathcal{G},k}), i = 1 \cdots N_F(N_G)$ be a facade or ground feature point. The association of $\mathbf{p}_i$ is the plane constructed by $\mathbf{q}_j, \mathbf{q}_m, \mathbf{q}_n$ in the last ground and facade feature points, which represent the closest point of $\bar{\mathbf{T}}_{k-1,k} \cdot \mathbf{p}_i$, the closest neighbor in the preceding and following scan lines to $\mathbf{q}_j$ and the closest neighbor in the same scan line to $\mathbf{q}_j$, respectively. The constraint equation is formulated as the point-to-plane distance:

$$
\begin{aligned}
r_{\mathcal{G},i} = r_{\mathcal{F},i} &= \mathbf{n}_j \cdot (\mathbf{T}_{k-1,k} \cdot \mathbf{p}_i), \\
\mathbf{n}_j &= \frac{(\mathbf{q}_j - \mathbf{q}_m) \times (\mathbf{q}_j - \mathbf{q}_n)}{\|(\mathbf{q}_j - \mathbf{q}_m) \times (\mathbf{q}_j - \mathbf{q}_n)\|}.
\end{aligned}
\tag{12}
$$

The $N_F \times 1$ facade feature error vector $\mathbf{r}_{\mathcal{F}}$ and the $N_G \times 1$ ground feature error vector $\mathbf{r}_{\mathcal{G}}$ are constructed by stacking all normalized facade and ground residuals (Line 8–9).

*III. Point-to-model intensity difference constraint.* The constraint equation is formulated as Equation (7). The $N_R \times 1$ intensity feature error vector $\mathbf{r}_{\mathcal{R}}$ is constructed by stacking all reflector features (Line 11).

### 3.4.2. Transformation Estimation

According to constraint models introduced above, the nonlinear least square (LS) function can be established for the transformation estimation (Line 12):

$$
\tilde{\mathbf{T}}_{k-1,k} = \underset{\mathbf{T}_{k-1,k}}{\arg\min} \left( \mathbf{r}_{\mathcal{G}}^{\top} \mathbf{r}_{\mathcal{G}} + \mathbf{r}_{\mathcal{F}}^{\top} \mathbf{r}_{\mathcal{F}} + \mathbf{r}_{\mathcal{E}}^{\top} \mathbf{r}_{\mathcal{E}} + \mathbf{r}_{\mathcal{R}}^{\top} \mathbf{r}_{\mathcal{R}} \right).
\tag{13}
$$

The *special Euclidean group* $\exp(\boldsymbol{\xi}_{k-1,k}^{\wedge}) = \mathbf{T}_{k-1,k}$ is implemented during the nonlinear optimization iteration. Then, $\mathbf{T}_{k-1,k}$ can be incrementally updated by:

$$
\boldsymbol{\xi}_{k-1,k} \leftarrow \boldsymbol{\xi}_{k-1,k} + \delta\boldsymbol{\xi}_{k-1,k}.
\tag{14}
$$

where

$$
\begin{aligned}
\delta\boldsymbol{\xi}_{k-1,k} &= \left(\mathbf{J}^{\top}\mathbf{J}\right)^{-1}\mathbf{J}^{\top}\mathbf{r}, \\
\mathbf{J} &= \left[ \mathbf{J}_{\mathcal{G},i}^{\top} \cdots \mathbf{J}_{\mathcal{F},i}^{\top} \cdots \mathbf{J}_{\mathcal{E},i}^{\top} \cdots \mathbf{J}_{\mathcal{R},i}^{\top} \right]^{\top}, \\
\mathbf{r} &= \left[ \mathbf{r}_{\mathcal{G},i}^{\top} \cdots \mathbf{r}_{\mathcal{F},i}^{\top} \cdots \mathbf{r}_{\mathcal{E},i}^{\top} \cdots \mathbf{r}_{\mathcal{R},i}^{\top} \right]^{\top}.
\end{aligned}
\tag{15}
$$

The Jacobian matrix of constraint equation with respect to $\boldsymbol{\xi}_{k-1,k}$ is denoted as $\mathbf{J}$. Matrix components are listed as follows.

$$
\begin{aligned}
\mathbf{J}_{\mathcal{G},i} &= \frac{\partial \mathbf{r}_{\mathcal{G},i}}{\partial \delta\boldsymbol{\xi}_{k-1,k}} = \mathbf{n}_{j,m,n}^{\top} \cdot \frac{\partial (\mathbf{T}_{k-1,k}\mathbf{p}_i)}{\partial \delta\boldsymbol{\xi}_{k-1,k}}, \\
\mathbf{J}_{\mathcal{F},i} &= \frac{\partial \mathbf{r}_{\mathcal{F},i}}{\partial \delta\boldsymbol{\xi}_{k-1,k}} = \mathbf{n}_{j,m,n}^{\top} \cdot \frac{\partial (\mathbf{T}_{k-1,k}\mathbf{p}_i)}{\partial \delta\boldsymbol{\xi}_{k-1,k}}, \\
\mathbf{J}_{\mathcal{E},i} &= \frac{\partial \mathbf{r}_{\mathcal{E},i}}{\partial \delta\boldsymbol{\xi}_{k-1,k}} \\
&= \frac{(\mathbf{v}_{j,m}^{\wedge}(\mathbf{T}_{k-1,k}\mathbf{p}_i))^{\top}}{\|\mathbf{v}_{j,m}^{\wedge}(\mathbf{T}_{k-1,k}\mathbf{p}_i)\|} \cdot \mathbf{v}_{j,m}^{\wedge} \cdot \frac{\partial (\mathbf{T}_{k-1,k}\mathbf{p}_i)}{\partial \delta\boldsymbol{\xi}_{k-1,k}}, \\
\mathbf{J}_{\mathcal{R},i} &= \frac{\partial \mathbf{r}_{\mathcal{R},i}}{\partial \delta\boldsymbol{\xi}_{k-1,k}} \\
&= \frac{\partial \boldsymbol{\phi}(\mathbf{T}_{k-1,k}\mathbf{p}_i)^{\top}}{\partial (\mathbf{T}_{k-1,k}\mathbf{p}_i)} \cdot \frac{\partial (\mathbf{T}_{k-1,k}\mathbf{p}_i)}{\partial \delta\boldsymbol{\xi}_{k-1,k}} \cdot \mathbf{c}_{\bar{\mathbf{q}}_i}.
\end{aligned}
\tag{16}
$$

### 3.5. LiDAR Mapping

There is always an inevitable error accumulation in the LiDAR odometry, resulting in a discrepancy $\Delta\mathbf{T}_k$ between the estimated and actual pose. In other words, the estimated transform from the LiDAR odometry thread is not the exact transform from the LiDAR frame $\{L\}$ to the world frame $\{W\}$ but from $\{L\}$ to the drifted world frame $\{W'\}$:

$$\mathbf{T}_{w,k} = \Delta\mathbf{T}_k\mathbf{T}_{w',k}. \tag{17}$$

One of the main tasks of the LiDAR mapping thread is optimizing the estimated pose from the LO thread by the scan-to-map registration (green S2M block in Figure 2). The other is managing the local static map (brown TVF and pink DOR blocks in Figure 2). The pseudocode is shown in Algorithm 2.

---

**Algorithm 2:** LiDAR Mapping

---

**Input:** Extracted feature points for registration $\mathcal{P}'_{\mathcal{G},k}$, $\mathcal{P}'_{\mathcal{F},k}$, $\mathcal{P}'_{\mathcal{E},k}$, $\mathcal{P}'_{\mathcal{R},k}$, feature points for mapping, $\mathcal{P}_{\mathcal{G},k}$, $\mathcal{P}_{\mathcal{F},k}$, $\mathcal{P}_{\mathcal{E},k}$, $\mathcal{P}_{\mathcal{R},k}$, estimated transform from LiDAR odometry $\mathbf{T}_{w',k}$, scan depth image $\mathcal{D}_k$ and labeled image $\mathcal{L}_k$

**Output:** refined pose $\mathbf{T}_{w,k}$, static scan points $\mathcal{P}_{s,k}$

1   *the transform drift $\Delta\mathbf{T}_k$ and scan keyframes can be loaded from the buffer;*

    `// Roughly transform the reckoned pose to the world frame`

2   $\bar{\mathbf{T}}_{w,k} \leftarrow \Delta\mathbf{T}_k \cdot \mathbf{T}_{w',k}$;

    `// Main`

3   **if** *skip a number of frames to keep system efficiency* **then**

       `// construct local points map`

4      $\mathcal{M}_k \leftarrow$ `searchSurroundKF`$(\bar{\mathbf{T}}_{w,k})$;

5      `temporalVoxelFilter`$(\mathcal{M}_k)$;

6      **for** *a number of iterations* **do**

         `// Find feature associations by parallel threads`

7          $\mathbf{r}_{\mathcal{G}}, \mathbf{J}_{\mathcal{G}} \leftarrow$ `GroundAssoc`$(\bar{\mathbf{T}}_{w,k}, \mathcal{P}'_{\mathcal{G},k}, \mathcal{M}_{G,k})$;

8          $\mathbf{r}_{\mathcal{F}}, \mathbf{J}_{\mathcal{F}} \leftarrow$ `FacadeAssoc`$(\bar{\mathbf{T}}_{w,k}, \mathcal{P}'_{\mathcal{F},k}, \mathcal{M}_{F,k})$;

9          $\mathbf{r}_{\mathcal{E}}, \mathbf{J}_{\mathcal{E}} \leftarrow$ `EdgeAssoc`$(\bar{\mathbf{T}}_{w,k}, \mathcal{P}'_{\mathcal{E},k}, \mathcal{M}_{E,k})$;

10        $\mathbf{r}_{\mathcal{R}}, \mathbf{J}_{\mathcal{R}} \leftarrow$ `ReflectAssoc`$(\bar{\mathbf{T}}_{w,k}, \mathcal{P}'_{\mathcal{R},k}, \mathcal{M}_{R,k})$;

         `// Update the estimated pose by the nonlinear optimization`

11        $\tilde{\mathbf{T}}_{w,k} \leftarrow$ `MultiMetricNLO`$(\mathbf{J}, \mathbf{r})$;

         `// Convergency`

12        *convergency* $\leftarrow$ `ConvergCond`$(\tilde{\mathbf{T}}_{w,k} \cdot \bar{\mathbf{T}}_{w,k}^{-1})$;

13        **if** *convergency* **then**

14          **break**;

15        **end**

16      **end**

       `// Update parameters`

17      $\mathbf{T}_{w,k} \leftarrow \tilde{\mathbf{T}}_{w,k}$;

18      $\mathbf{T}_{w',w}^{-1} = \mathbf{T}_{w,k} \cdot \bar{\mathbf{T}}_{w,k}$;

       `// Update the local feature map`

19      `DownsizeFilter`$(\mathcal{P}_{\mathcal{G},k}, \mathcal{P}_{\mathcal{F},k}, \mathcal{P}_{\mathcal{E},k}, \mathcal{P}_{\mathcal{R},k})$;

20      $\mathcal{P}_{s,k} \leftarrow$ `DORFilter`$(\mathcal{P}_k, \mathcal{M}_k, \mathcal{D}_k, \mathcal{L}_k, \mathbf{T}_{w,k})$;

21      `InsertAsKF`$(\mathcal{P}_{s,k}, \mathbf{T}_{w,k})$;

22 **end**

---

#### 3.5.1. Local Feature Map Construction

In this paper, the pose-based local feature map construction scheme is applied. In particular, the pose prediction $\bar{\mathbf{T}}_{w,k}$ is calculated by Equation (17) under the assumption that the drift between $\Delta\mathbf{T}_k$ and $\Delta\mathbf{T}_{k-1}$ is tiny (Line 2). Feature points scanned in the vicinity of $\bar{\mathbf{T}}_{w,k}$ are merged (Line 4) and filtered (Line 5) to construct the local map $\mathcal{M}_k$. Let $\Gamma(\cdot)$

denote the filter and $n \in N$ denote timestamps of surrounding scans. The local map is built by:

$$\mathcal{M}_k = \Gamma \left( \sum_{n \in N} \mathbf{T}_{w,n} \cdot \mathcal{P}_{s,n} \right). \tag{18}$$

The conventional voxel-based downsample filter voxelizes the point cloud and retains one point for each voxel. The coordinate of the retained point is averaged by all points in the same voxel. However, for the point intensity, averaging may cause the loss of similarity between consecutive scans. To maintain the local characteristic of the point intensity, we utilize the temporal information to improve the voxel-based downsample filter. In the TVF, a temporal window is set for the intensity average. Specifically, the coordinate of the downsampled point is still the mean of all points in the voxel, but the intensity is the mean of points in the temporal window, i.e., $|t_k - t_n| < Th_t$, where $t_k$ and $t_n$ represent timestamps of the current scan and selected point, respectively.

### 3.5.2. Mapping Update

The categorized features are jointly registered with feature maps in the same way as in the LiDAR odometry module. The low-drift pose transform $\mathbf{T}_{w,k}$ can be estimated by scan-to-map alignment (Line 7–11). Since the distribution of feature points in the local map is disordered, point neighbors cannot be directly indexed through the scan line number. Accordingly, the K-D tree is utilized for nearest point searching and the PCA algorithm calculates norms and primary directions of neighboring points.

Finally, the obtained $\mathbf{T}_{w,k}$ is fed to the DOR filter to filter out dynamic points in the current scan. Only static points $\mathcal{P}_{s,k}$ are retained in the local feature map list (Line 20–21). Moreover, the odometry reference drift is also updated by Equation (17), i.e., $\Delta\mathbf{T}_k = \mathbf{T}_{w,k}\mathbf{T}_{w',k}^{-1}$ (Line 18).

### 4. Results

In this section, the proposed InTEn-LOAM is evaluated qualitatively and quantitatively on both simulated and real-world datasets, covering various outdoor scenes. We first test the feasibility of each functional module, i.e., feature extraction, intensity-based scan registration and dynamic point removal. Then, we conduct a comprehensive evaluation for InTEn-LOAM in terms of positioning accuracy and constructed map quality. We run the proposed LO system on a laptop computer with 1.8 GHz quad cores and 4 Gib memory, on top of the robot operating system (ROS) in Linux.

The simulated test environment was built based on the challenging scene provided by the DARPA Subterranean (SubT) Challenge (https://github.com/osrf/subt, accessed on 1 October 2022). We simulated a 1000 m long straight mine tunnel (see Figure 7b) with smooth walls and reflective signs that are alternatively posted on both sides of the tunnel at 30 m intervals. Physical parameters of the simulated car, such as ground friction, sensor temperature and humidity are consistent with reality to the greatest extent. A 16-scanline LiDAR is mounted on the top of the car. Transform groundtruths were exported at 100 Hz. The real-world dataset was collected by an autonomous driving car with a 32-scanline LiDAR (see Figure 7a) in the autonomous driving test field, where a 150 m long straight tunnel exists. Moreover, the KITTI odometry benchmark (http://www.cvlibs.net/datasets/kitti/eval_odometry.php, accessed on 1 October 2022) was also utilized to compare with other state-of-the-art LO solutions.

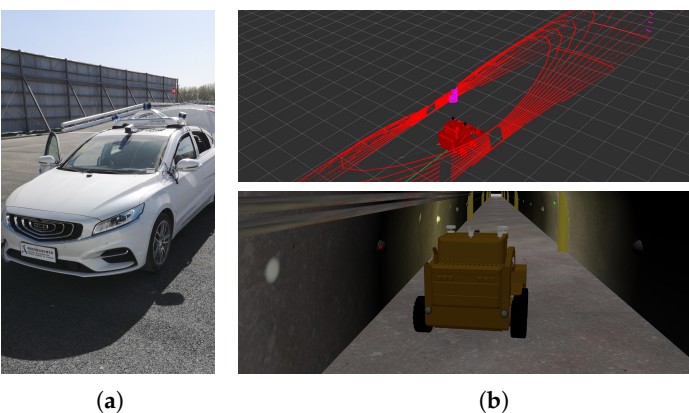

(**a**)                                  (**b**)

**Figure 7.** Dataset sampling platform. (**a**) Autonomous driving car; (**b**) Simulated mine car and scan example. Magenta lazer points are reflected from brown signs in the simulated environment.

*4.1. Functional module Experiments*

4.1.1. Feature Extraction Test

We validated the feature extraction module on the real-world dataset. In the test, we set the edge feature extraction threshold as $Th_E = 0.3$, the facade feature extraction threshold as $Th_E = 0.1$ and the intensity difference threshold as $Th_{\Delta I} = 80$ and partitioned the intensity image into $16 \times 4$ blocks.

Figure 8 shows feature extraction results. It can be seen that edges, planes and reflectors can be correctly extracted in various road conditions. With the effect of ground segmentation, breakpoints on the ground (see orange box region in Figure 8a) are correctly marked as plane, avoiding the issue that breakpoints are wrongly marked as edge features due to their large roughness values. In the urban city scene, conspicuous intensity features can be easily found, such as landmarks and traffic lights (see Figure 8b). Though there are many plane features in the tunnel, few valid edge features can be extracted (see Figure 8c). In addition, sparse and scattered plant points with large roughness values (see orange box region in Figure 8d) are filtered as outliers with the help of object clustering.

According to the above results, some conclusions can be drawn: (1) The number of plane features is always much greater than that of edge features, especially in open areas, which may cause the issue of constraint-unbalance during the multi-metric nonlinear optimization. (2) Static reflector features widely exist in real-world environments, which are useful for the feature-based scan alignment and should not be ignored. (3) The adaptive intensity feature extraction approach makes it possible to manually add reflective targets in feature-degraded environments.

4.1.2. Feature Ablation Test

We validated the intensity-based scan registration method on the simulated dataset. To highlight the effect of intensity features in the scan registration, we quantitatively evaluated the relative accuracy of the proposed intensity-based scan registration method and compared the result with prevalent geometric-based scan registration methods, i.e., edge and surface feature registration of LOAM [8], multi-metric registration of MULLS [27] and NDT of HDL-Graph-SLAM [13]. The evaluation used the simulated tunnel dataset, which is a typical geometric-degraded environment. The measure used to evaluate the accuracy of scan registration is the relative transformation error. In particular, differences between the groundtruth $\mathbf{T}_{k+1,k}^{GT}$ and the estimated relative transformation $\mathbf{T}_{k+1,k}$ are calculated and represented as an error vector, i.e., $\mathbf{r}_k = \text{vec}(\mathbf{T}_{k+1,k}^{GT} \cdot \mathbf{T}_{k+1,k}^{-1})$. The norms of translational and rotational parts of $\mathbf{r}_k$ are illustrated in Figure 9. Note that the result of intensity-based registration only utilizes measurements from the intensity channel of the lazer scan, i.e., only intensity features are used for intensity matching.

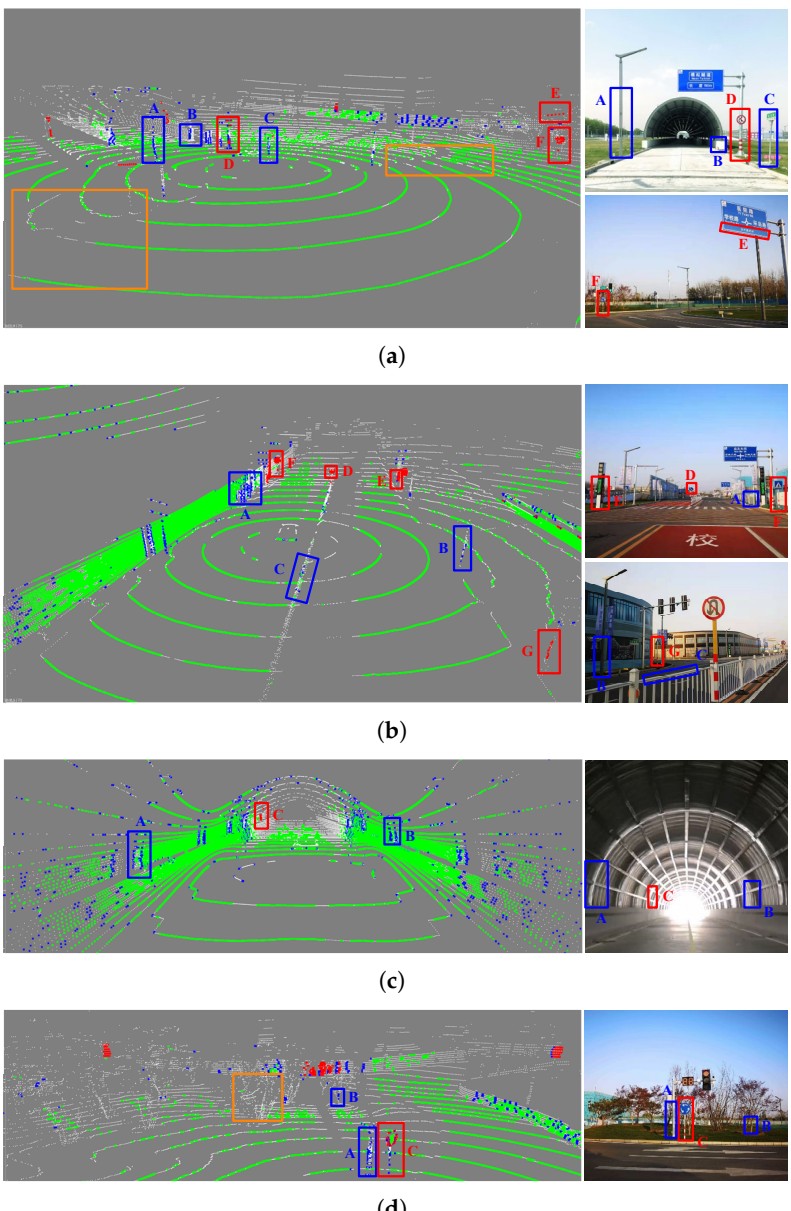

**Figure 8.** Feature extraction results in different scenes. (**a**) Open road; (**b**) City avenue; (**c**) Long straight tunnel; (**d**) Roadside green belt. (plane, reflector, edge and raw scan points). Objects in the real-world scenes and their counterparts in lazer scans are indicated by boxes (reflector features, edge features, some special areas).

The figures show that all four rotation errors of different approaches are less than $0.01°$, while errors of InTEn-LOAM and MULLS are less than $0.001°$. This demonstrates that lazer points from the tunnel wall and ground enable one to provide sufficient geometric constraints for the accuracy of relative attitude estimation. However, there are significant differences in relative translation errors (RTEs). The intensity-based scan registration achieves the best RTE (less than 0.02 m), which is much better than the feature-based sort of LOAM and NDT of HDL-Graph-SLAM (0.4 m and 0.1 m) and better than the intensity-based weighting of MULLS (0.05). The result proves the correctness and feasibility of the proposed intensity-based approach under the premise of sufficient intensity features. It also reflects the necessity of fusing reflectance information of points in poorly structured environments.

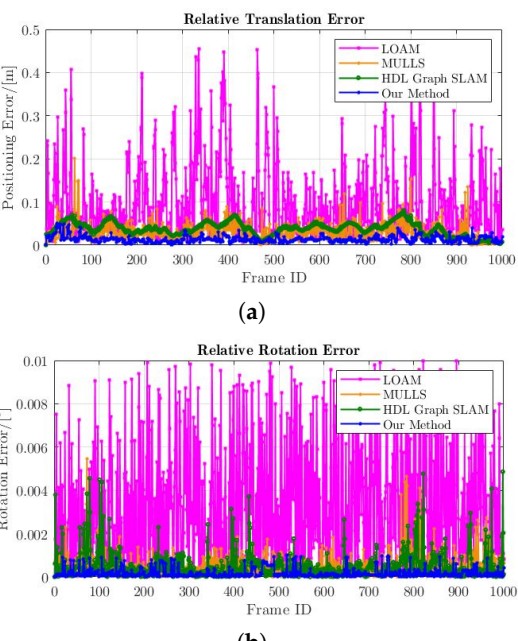

**Figure 9.** Relative error plots. (**a**) Relative translation error curves; (**b**) Relative rotation error curves. (NDT of HDL-Graph-SLAM, feature-based registration approach of LOAM, the proposed intensity-based registration approach).

### 4.1.3. Dynamic Object Removal

We validated the DOR module on Seq.07 and 10 of the KITTI odometry dataset. The test result was evaluated via the qualitative evaluation method, i.e., marking dynamic points for each scan frame and qualitatively judging the accuracy of the dynamic object segmentation according to the actual targets in the real world the dynamic points correspond to.

Figure 10 exhibits DOR examples for a single frame of lazer scan at typical urban driving scenes. It can be seen that dynamic objects, such as vehicles crossing the intersection, vehicles and pedestrians traveling in front of/behind the data collection car, can be correctly segmented via the proposed DOR approach no matter whether the sampling vehicle is stationary or in motion. Figure 11 shows constructed maps at two representative areas, i.e., intersection and busy road. Maps were incrementally built by LOAM (without DOR) and InTEn-LOAM (with DOR) methods. We can figure out from the figure that the map built by InTEn-LOAM is better since the DOR module effectively filters out dynamic points to help to accumulate a purely static points map. In contrast, the map constructed via LOAM has a large amount of 'ghost points' increasing the possibility of erroneous point matching.

In general, the above results prove that the DOR method proposed in this paper has the ability to segment dynamic objects for a scan frame correctly. However, it also has some shortcomings. For instance, (1) the proposed comparison-based DOR filter is sensitive to the quality of lazer scan and the density of the local points map, causing the omission or mis-marking of some dynamic points (see the green box at the top of Figure 10a and the red rectangle box at the bottom of Figure 10b); (2) dynamic points in the first frame of the scan cannot be marked using the proposed approach (see the red rectangle box in the top of Figure 10b).

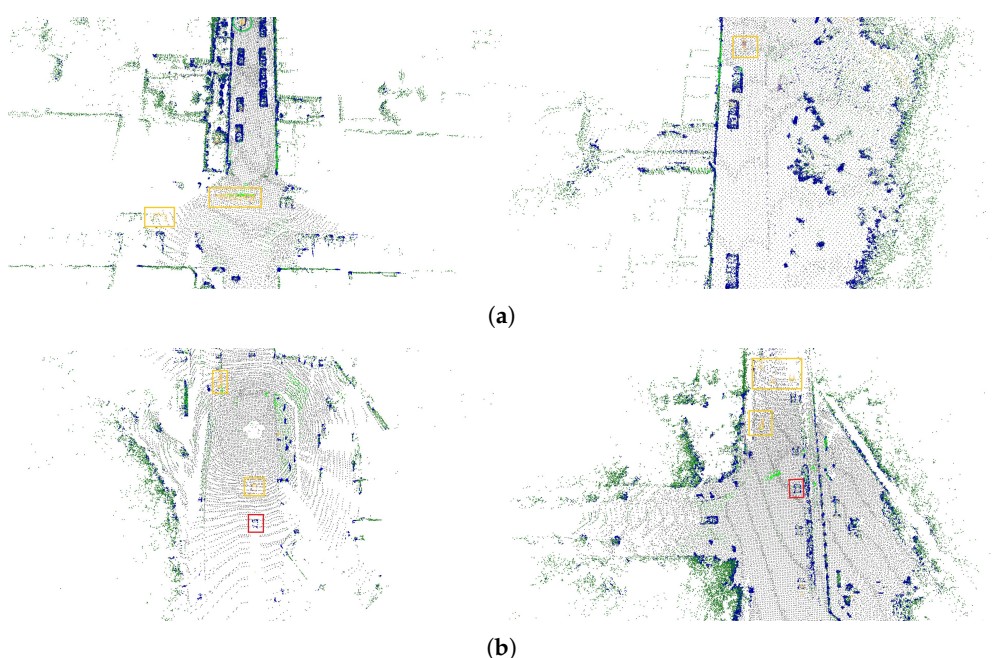

(**a**)

(**b**)

**Figure 10.** DOR examples for a single frame of lazer scan. (**a**) Seq.07. Vehicles crossing the intersection when the data collection vehicle stops and waits for the traffic light (**left**); The cyclist traveling in the opposite direction when the data collection vehicle is driving along the road (**right**). (**b**) Seq.10. Followers behind the data collection vehicle as it travels down the highway at high speed (**top**); Vehicles driving in the opposite direction and in front of the data collection vehicle when it slows down (**bottom**). (facade, ground, edge and dynamics for points true positive, false positive and true negative for dynamic segmentation boxes.)

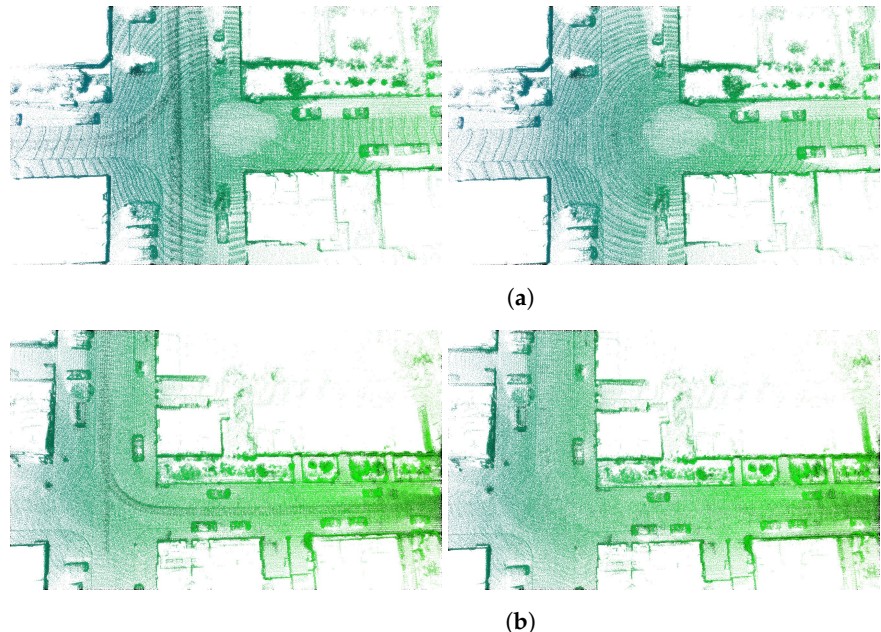

(**a**)

(**b**)

**Figure 11.** Comparison between local maps of LOAM and InTEn-LOAM. (**a**) Map at the intersection; (**b**) Map at the busy road. In each subfigure, the top represents the map of LOAM without DOR, while the bottom represents the map of InTEn-LOAM with DOR.

## 4.2. Pose Transform Estimation Accuracy

### 4.2.1. KITTI Dataset

The quantitative evaluations were conducted on the KITTI odometry dataset, which is composed of 11 sequences of lazer scans captured by a Velodyne HDL-64E LiDAR with

GPS/INS groundtruth poses. We followed the odometry evaluation criterion from [45] and used the average relative translation and rotation errors (RTEs and RREs) within a certain distance range for the accuracy evaluation. The performance of the proposed InTEn-LOAM and the other six state-of-the-art LiDAR odometry solutions whose results are taken from their original papers are reported in Table 1. Only LO state-of-the-art items without the back-end optimization or semantic segmentation or other sensors' fusion are compared to ensure fairness. Plots of average RTE/RRE over fixed lengths are exhibited in Figure 12. Note that all comparison methods did not incorporate the loop closure module for more objective accuracy comparison. Moreover, an intrinsic angle correction of 0.2° is applied to KITTI raw scan data for better performance [27].

**Table 1.** Quantitative evaluation and comparison on KITTI dataset. Note: All errors are represented as average RTE[%]/RRE[∘/100 m]. Red and blue fonts denote the first and second place, respectively. Denotations: U: Urban road; H: Highway; C: Country road.

| Method | #00U | #01H | #02C | #03C | #04C | #05C | #06U | #07U | #08U | #09C | #10C | Avg. | Time [s]/Frame |
|---|---|---|---|---|---|---|---|---|---|---|---|---|---|
| LOAM | 0.78/- | 1.43/- | 0.92/- | 0.86/- | 0.71/- | 0.57/- | 0.65/- | 0.63/- | 1.12/- | 0.77/- | 0.79/- | 0.84/- | 0.10 |
| IMLS-SLAM | 0.50/- | 0.82/- | 0.53/- | 0.68/- | 0.33/- | 0.32/- | 0.33/- | 0.33/- | 0.80/- | 0.55/- | 0.53/- | 0.57/- | 1.25 |
| MC2SLAM | 0.51/- | 0.79/- | 0.54/- | 0.65/- | 0.44/- | 0.27/- | 0.31/- | 0.34/- | 0.84/- | 0.46/- | 0.52/- | 0.56/- | 0.10 |
| SuMa | 0.70/0.30 | 1.70/0.50 | 1.10/0.40 | 0.70/0.50 | 0.40/0.30 | 0.40/0.20 | 0.50/0.30 | 0.70/0.60 | 1.00/0.40 | 0.50/0.30 | 0.70/0.30 | 0.70/0.30 | 0.07 |
| LO-Net | 0.78/0.42 | 1.42/0.40 | 1.01/0.45 | 0.73/0.59 | 0.56/0.54 | 0.62/0.35 | 0.55/0.35 | 0.56/0.45 | 1.08/0.43 | 0.77/0.38 | 0.92/0.41 | 0.83/0.42 | 0.10 |
| MULLS-LO | 0.51/0.18 | 0.62/0.09 | 0.55/0.17 | 0.61/0.22 | 0.35/0.08 | 0.28/0.17 | 0.24/0.11 | 0.29/0.18 | 0.80/0.25 | 0.49/0.15 | 0.61/0.19 | 0.49/0.16 | 0.08 |
| InTEn-LOAM | 0.51/0.21 | 0.63/0.35 | 0.54/0.28 | 0.63/0.33 | 0.37/0.31 | 0.36/0.25 | 0.24/0.11 | 0.34/0.31 | 0.71/0.29 | 0.48/0.19 | 0.45/0.21 | 0.54/0.26 | 0.09 |

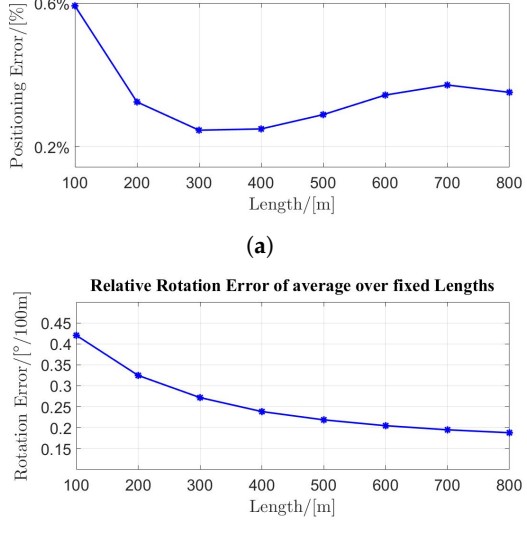

**(a)**

**(b)**

**Figure 12.** The average RTE and RRE of InTEn-LOAM over fixed lengths. (**a**) RTE; (**b**) RRE.

Figure 12 demonstrates that accuracies in different length ranges are stable and the maximums of average RTE and RRE are less than 0.32% and 0.22°/100 m. It also can be seen from the table that the average RTE and RRE of InTEn-LOAM are 0.54% and 0.26°/100 m, which outperforms the LOAM accuracy of 0.84%. The comprehensive comparison shows that InTEn-LOAM is superior or equal to the current state-of-the-art LO methods. Although the result of MULLS is slightly better than that of InTEn-LOAM, the contribution of InTEn-LOAM is significant, considering its excellent performance in long straight tunnels with reflective markers. InTEn-LOAM costs around 90 ms per frame of scan with about 3 k and 30 k feature points in the current scan points and local map points, respectively. Accordingly, the proposed LO method is able to operate faster than 10 Hz on average for all KITTI odometry sequences and achieve real-time performance.

For in-depth analysis, three representative sequences, i.e., Seq.00, 01 and 05, were selected. Seq.00 is an urban road dataset with the longest traveling distance, in which

big and small loop closures are included, while geometric features are extremely rich. Consequently, the sequence is suitable for visualizing the trajectory drift of InTEn-LOAM. Seq.01 is a highway dataset with the fastest driving speed. Due to the lack of geometric features in the highway neighborhood, it is the most challenging sequence in the KITTI odometry dataset. Seq.05 is a country road sequence with great variation in elevation and rich structured features.

For Seq.01, it can be seen from Figure 13c that areas with landmarks are indicated by blue bounding boxes, while magenta boxes highlight road signs on the roadside. The drift of the estimated trajectory of Seq.01 by InTEn-LOAM is quite small (see Figure 13d), which reflects that the roadside guideposts can be utilized as reflector features since their high-reflective surfaces are conducive to improving the LO performance in such geometric-sparse highway environments. The result also proves that the proposed InTEn-LOAM is capable of adaptively mining and fully exploiting the geometric and intensity features in surrounding environments, which ensures the LO system can accurately and robustly estimate the vehicle pose even in some challenging scenarios. In terms of Seq.00 and 05, both two point cloud maps show excellent consistency in the small loop closure areas (see blue bound regions in Figure 13a,e), which indicates that InTEn-LOAM has good local consistency. However, in large-scale loop closure areas, such as the endpoint, the global trajectory drifts incur a stratification issue in point cloud maps (see red bound regions in Figure 13a,e), which are especially significant in the vertical direction (see plane trajectory plots in Figure 13b,f). This phenomenon is due to constraints in the z-direction being insufficient in comparison with other directions in the state space since only ground features provide constraints for the z-direction during the point cloud alignment.

### 4.2.2. Autonomous Driving Dataset

The other quantitative evaluation test was conducted on the autonomous driving field dataset, the groundtruth of which refers to the trajectory output of the onboard positioning and orientation system (POS). There is a 150 m long tunnel in the data acquisition environment, which is extremely challenging for most LO systems. The root mean square errors (RMSEs) of horizontal position and yaw angle were used as indicators for the absolute state accuracy. LOAM, MULLS and HDL-Graph-SLAM were utilized as control groups, whose results are listed in Table 2.

Both LOAM and HDL-Graph-SLAM failed to localize the vehicle with 34.654 m and 141.498 m positional errors, respectively. MULLS and the proposed InTEn-LOAM are still able to function properly with 2.664 m and 7.043 m of positioning error and 0.476° and 1.403° of heading error within the path range of 1.5 km. To further investigate the causes of this result, we plotted the cumulative distribution of absolute errors and horizontal trajectories of three LO systems, as shown in Figure 14.

From the trajectory plot, we can see that the overall trajectory drift of InTEn-LOAM and MULLS are relatively small, indicating that these two approaches can accurately localize the vehicle in this challenging scene by incorporating intensity features into the point cloud registration and using intensity information for the feature weighting. The estimated position of LO inevitably suffers from error accumulation which is the culprit causing trajectory drift. It can be seen from the cumulative distribution of absolute errors that the absolute positioning error of InTEn-LOAM is no more than 10 m and the attitude error is no more than 1.5°. The overall trends of rotational errors of the other three systems are consistent with that of InTEn-LOAM. Results in Table 2 also verify that their rotation errors are similar. The cumulative distribution curves of absolute positioning errors of LOAM and HDL-Graph-SLAM do not exhibit smooth growth but a steep increase in some intervals. The phenomenon reflects the existence of anomalous registration in these regions, which is consistent with the fact that the scan registration-based motion estimation in the tunnel is degraded. MULLS, which incorporates intensity measures by feature constraint weighting, presents a smooth curve similar to InTEn-LOAM. However, the absolute errors of positioning (no more than 19 m) and heading (no more than 2.5°) are

both larger than those of our proposed LO system. We also plotted the RTE and RRE of all four approaches (see Figure 15). It can be seen that the differences in the RRE between the four systems are small, representing that the heading estimations of all these LO systems are not deteriorated in the geometric-degraded long straight tunnel. In contrast, the RPEs are quite different. Both LOAM and HDL-Graph-SLAM suffer from serious scan registration drifts, while MULLS and InTEn-LOAM are able to position normally and achieve very close relative accuracy.

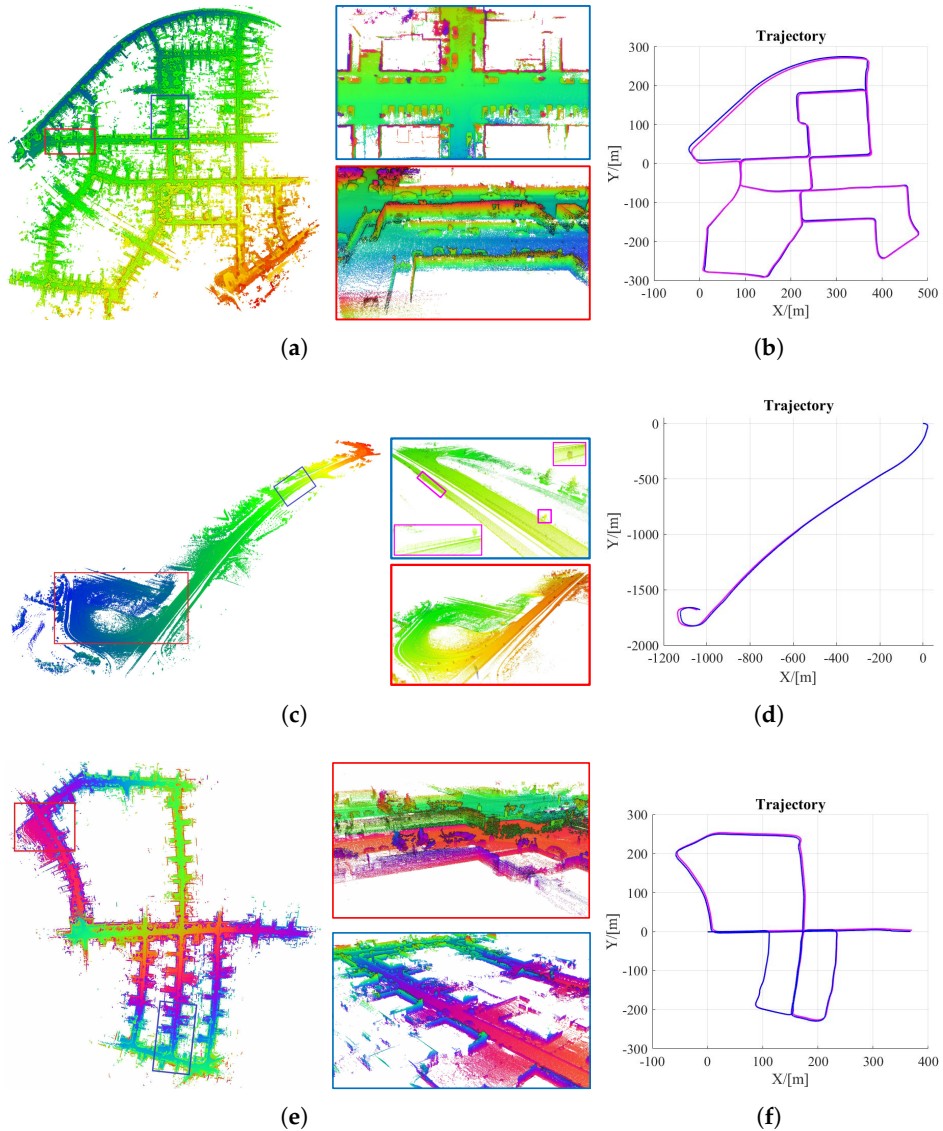

(**a**)        (**b**)

(**c**)        (**d**)

(**e**)        (**f**)

**Figure 13.** Constructed points maps with details and estimated trajectories. (**a**,**c**,**e**) maps of Seq.00, 01 and 05; (**b**,**d**,**f**) trajectories of Seq.00, 01 and 05 (groundtruths and InTEn-LOAM).

**Table 2.** Quantitative evaluation and comparison on autonomous driving field dataset.

| Method | Positioning Error (m) | | | Heading Error (°) |
|---|---|---|---|---|
| | *x* | *y* | Horizontal | Yaw |
| LOAM | 29.478 | 18.220 | 34.654 | 1.586 |
| HDL-Graph-SLAM | 119.756 | 75.368 | 141.498 | 2.408 |
| MULLS-LO | 4.133 | 5.705 | 7.043 | 1.403 |
| InTEn-LOAM | 1.851 | 1.917 | 2.664 | 0.476 |

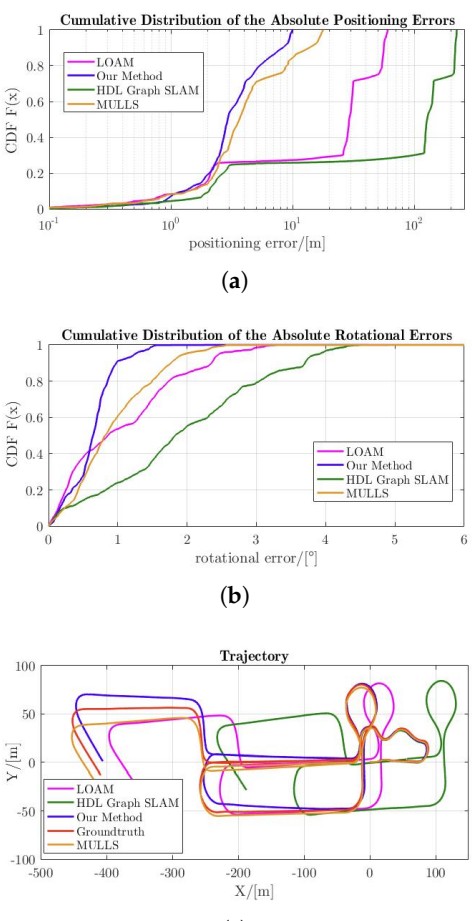

**Figure 14.** Cumulative distributions of absolute state errors and estimated trajectories. (**a**) Cumulative distributions of the absolute positioning errors; (**b**) Cumulative distributions of the absolute rotational errors; (**c**) Estimated trajectories. (InTEn-LOAM, LOAM, HDL-Graph-SLAM, groundtruth).

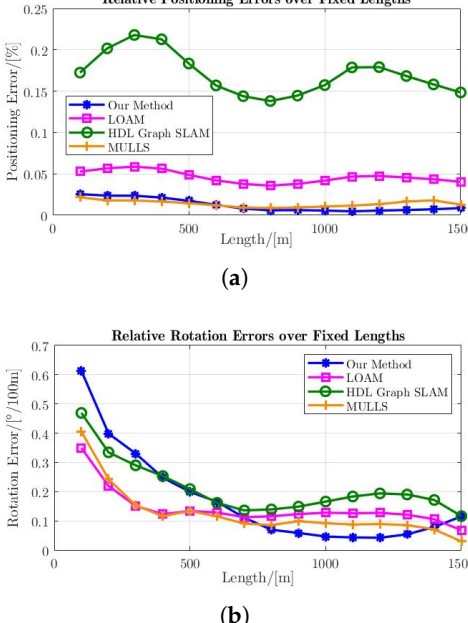

**Figure 15.** The average RTE and RRE of LO systems over fixed lengths. (**a**) RTE; (**b**) RRE.

### 4.3. Point Cloud Map Quality

4.3.1. Large-Scale Urban Scenario

The qualitative evaluations were conducted by intuitively comparing the map constructed by InTEn-LOAM with the reference map. The reference map is built by merging each frame of the lazer scan using their groundtruth poses. Maps of Seq.06 and 10 are displayed in Figures 16 and 17, which are the urban scenario with trajectory loops and the country road scenario without loop, respectively.

Although the groundtruth is the post-processing result of POS and its absolute accuracy reaches centimeter-level, the directly merged points map is blurred in the local view. By contrast, maps built by InTEn-LOAM have better local consistency, and various small targets, such as trees, vehicles, fences, etc., can be clearly distinguished from the points map. The above results prove that the relative accuracy of InTEn-LOAM outperforms that of the GPS/INS post-processing solution, which is very critical for the mapping tasks.

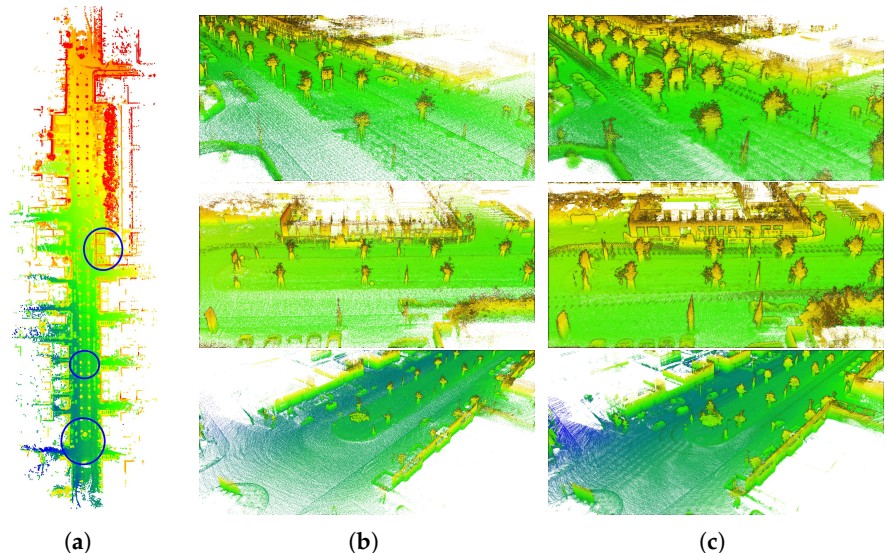

(**a**) (**b**) (**c**)

**Figure 16.** InTEn-LOAM's map result on urban scenario (KITTI seq.06): (**a**) overview, (**b**) map in detail of circled areas, (**c**) reference map comparison.

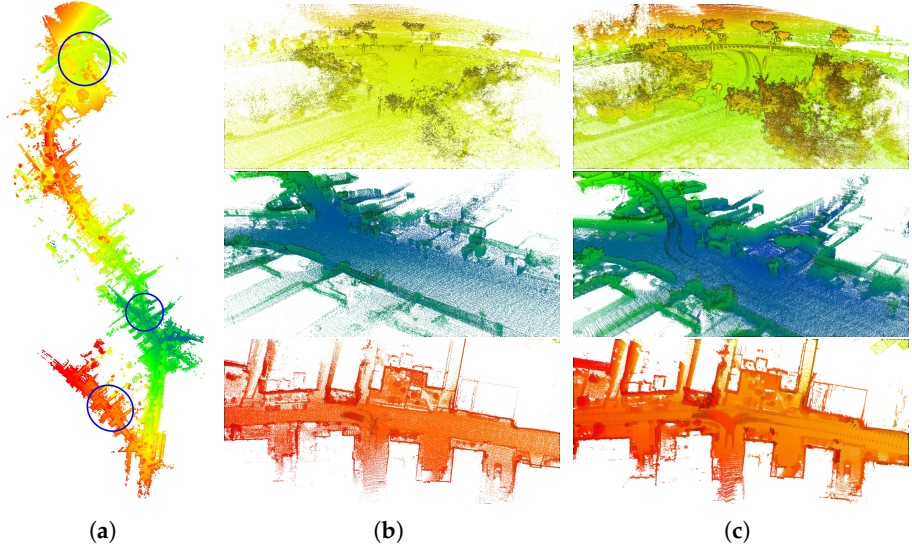

(**a**) (**b**) (**c**)

**Figure 17.** InTEn-LOAM's map result on country scenario (KITTI seq.10): (**a**) overview, (**b**) map in detail of circled areas, (**c**) reference map comparison.

#### 4.3.2. Long Straight Tunnel Scenario

The second qualitative evaluation test was conducted on the autonomous driving field dataset. There is a 150 m long straight tunnel in which we alternately posted some reflective signs on sidewalls to manually add some intensity features in such registration-degraded scenario. Maps of InTEn-LOAM, MULLS, LOAM and HDL-Graph-SLAM are shown in Figure 18. It intuitively shows that both LOAM and HDL-Graph-SLAM present different degrees of scan registration degradation, while the proposed InTEn-LOAM achieves correct motion estimation by jointly utilizing both sparse geometric and intensity features, as shown in Figure 18. Although MULLS is also able to build a correct map since it utilizes intensity information to re-weight geometric feature constraints during the registration iteration, its accuracies of both pose estimation and mapping are inferior to the proposed LO system.

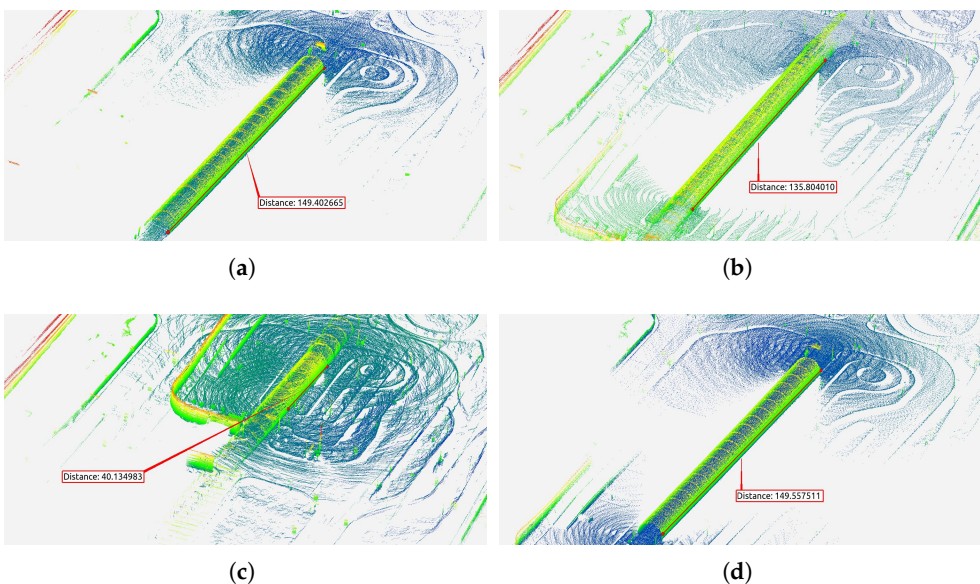

|     |     |
|:---:|:---:|
| (**a**) | (**b**) |
| (**c**) | (**d**) |

**Figure 18.** LO systems' map results on autonomous driving field dataset in the tunnel region. (**a**) InTEn-LOAM, (**b**) LOAM, (**c**) HDL-Graph-SLAM, (**d**) MULLS.

In addition, we constructed the complete point cloud map for the test field using InTEn-LOAM and compared the result with the local remote sensing image, as shown in Figure 19. It can be seen that the consistency between the constructed point cloud map and regional remote sensing image is good, qualitatively reflecting that the proposed InTEn-LOAM has excellent localization and mapping capability without error accumulation in the around 2 km long exploration journey.

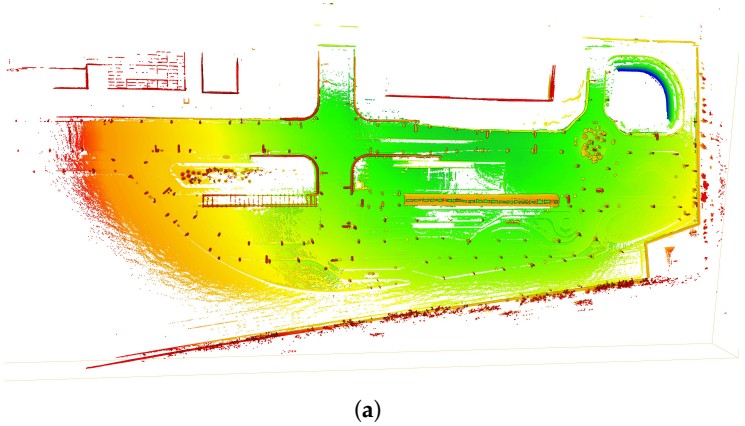

(**a**)

**Figure 19.** *Cont.*

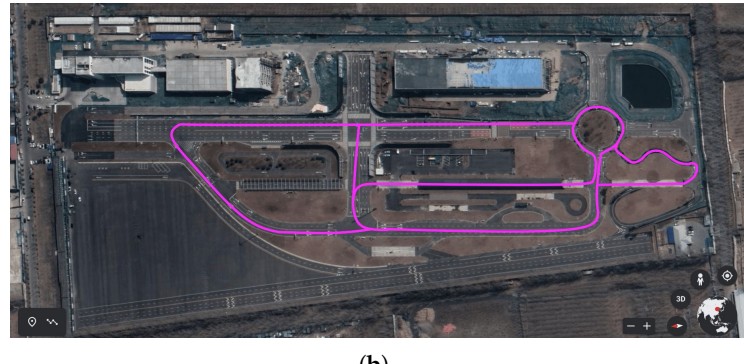

(**b**)

**Figure 19.** InTEn-LOAM's map result on autonomous driving field dataset. (**a**) the constructed point cloud map, (**b**) local remote sensing image and estimated trajectory.

### 5. Discussion

In this work, we present a LiDAR-only odometry and mapping solution named InTEn-LOAM to cope with some challenging issues, i.e., dynamic environments, intensity channel incorporation and feature degraded environments. The efficient and adaptive image-based feature extraction method, the temporal-based dynamic removal method and the novel intensity-based scan registration approach are proposed and all of them are utilized to improve the performance of LOAM. The proposed system is evaluated on both simulated and real-world datasets. Results show that InTEn-LOAM achieves similar or better accuracy in comparison with the state-of-the-art LO solutions in normal environments and outperforms them in challenging scenarios, such as long straight tunnels. Since the LiDAR-only method cannot adapt to aggressive motion, our future work involves developing an IMU/LiDAR tightly coupled method to escalate the robustness of motion estimation.

**Author Contributions:** S.L. conceived the methodology, wrote the manuscript, coded the software and performed the experiments; B.T. gave guidance and provided the test data resources; X.Z., J.G., W.Y. and G.L. reviewed the paper and gave constructive advice. All authors have read and agreed to the published version of the manuscript.

**Funding:** This research was funded by the National Natural Science Foundation of China (No.42201501 and No.42071454) and the Key-Area Research and Development Program of Guangdong Province (No.2020B0909050001 and No.2020B090921003).

**Data Availability Statement:** Not applicable.

**Conflicts of Interest:** The authors declare no conflict of interest.

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
