# Peer review of "InTEn-LOAM: Intensity and Temporal Enhanced LiDAR Odometry and Mapping"

_remotesensing, doi:10.3390/rs15010242_

Round 1

Reviewer 1 Report

Comments to the Author

The paper titled "InTEn-LOAM: Intensity and Temporal Enhanced LiDAR Odometry and Mapping", investigates in detail the effects of to present a low-drift, robust LIDAR odometry and mapping method inteni - loam, which takes full advantage of the geometric, intensity and temporal characteristics of laser scanning. The accuracy is shown to be close to or better than state-of-the-art in normal driving scenarios and better than geometry-based LO in unstructured environments.The paper has some novelties but should be carefully edited and improved.

 1. First and foremost, is the specific content of this work two aspects of method innovation and experimental verification? At first glance, it appears to contain both, but there isn't enough detail to appreciate that it is both. Regarding the abstract, these two aspects should be considered, and I recommend it to grab the reader's attention at a glance.

 2. At the end of this section, the author clearly highlights the innovations of this article. These four points include two methods, one filter element and experimental comparison respectively. In my opinion, this part not only makes me a little confused, but as a qualified research article, it must summarize the highlight highly. Method innovation and device innovation should be separated, and readers should have a clearer understanding of the innovation points of this paper.

 3. In the part of principle and model, the author gives a lot of principle explanation and model establishment, which is worthy of affirmation, but the content of this part is too miscellaneous, the establishment of the third level title will undoubtedly increase the length of the article. As far as I know the remote sensing is usually around page 19. Even if this part is what you highlight, the author should arrange the logic of this part to make it easier to read.

 4. The results of the comparison part of the work seems to be striking, the author compares the proposed system scheme with several previous systems respectively, which is worthy of affirmation. Despite the content, I see several grammatical and technical errors here. So I believe the same problem exists throughout the manuscript. Therefore, I hope the authors can carefully check and modify the vocabulary and grammar in the article.

 5. Minor suggestion:The discussion part needs to be consistent with the previous text, and it is suggested to enrich this part.

  To conclude, I think the manuscript meets the standard of remote sensing in terms of innovation. And, this work is undoubtedly a success! However, I hope that the author will systematically revise the full writing narrative, which will again improve the quality of the presentation of the manuscript.

Author Response

The paper titled "InTEn-LOAM: Intensity and Temporal Enhanced LiDAR Odometry and Mapping", investigates in detail the effects of to present a low-drift, robust LIDAR odometry and mapping method inten-loam, which takes full advantage of the geometric, intensity and temporal characteristics of laser scanning. The accuracy is shown to be close to or better than state-of-the-art in normal driving scenarios and better than geometry-based LO in unstructured environments. The paper has some novelties but should be carefully edited and improved.
AR: We appreciate the reviewer's detailed review and recognition of our work.
Comment 1: First and foremost, is the specific content of this work two aspects of method innovation and experimental verification? At first glance, it appears to contain both, but there isn't enough detail to appreciate that it is both. Regarding the abstract, these two aspects should be considered, and I recommend it to grab the reader's attention at a glance.
AR: Thanks for the reviewer's suggestion. We have reorganized the abstract to make it more clear. The changes are marked as blue in the revised manuscript.
Comment 2: At the end of this section, the author clearly highlights the innovations of this article. These four points include two methods, one filter element and experimental comparison respectively. In my opinion, this part not only makes me a little confused, but as a qualified research article, it must summarize the highlight highly. Method innovation and device innovation should be separated, and readers should have a clearer understanding of the innovation points of this paper.
AR: Thanks for the reviewer's suggestion. The main contributions contain innovative points extraction, filtering, and registration approaches, as well as extensive experimental tests. Therefore, we summarized the contributions as four-fold in deltails.
Comment 3: In the part of principle and model, the author gives a lot of principle explanation and model establishment, which is worthy of affirmation, but the content of this part is too miscellaneous, the establishment of the third level title will undoubtedly increase the length of the article. As far as I know the remote sensing is usually around page 19. Even if this part is what you highlight, the author should arrange the logic of this part to make it easier to read.
AR: We appreciate the reviewer's constructive suggestion. We have edited the manuscript to make it clearer.We did consider making it short for readability but found it's hard to explain the detail clearly. Therefore, we used the thrid level titles to show logic. The proposed LO system contains many innovative modules from feature extraction to laser sweep registration. To ensure the integrity of the whole LO system, we retain the third level tiltes though they increase the length of the paper. 
Comment 4: The results of the comparison part of the work seems to be striking, the author compares the proposed system scheme with several previous systems respectively, which is worthy of affirmation. Despite the content, I see several grammatical and technical errors here. So I believe the same problem exists throughout the manuscript. Therefore, I hope the authors can carefully check and modify the vocabulary and grammar in the article.
AR: Thanks for the reviewer's suggestion. We have revised all typos and mistakes in the paper. The changes are marked as blue in the revised manuscript.
Comment 5: Minor suggestion: The discussion part needs to be consistent with the previous text, and it is suggested to enrich this part.
AR: Thanks for the reviewer's suggestion. We have revised the Sect.V Discussion to keep it consistent with previous statement.
To conclude, I think the manuscript meets the standard of remote sensing in terms of innovation. And, this work is undoubtedly a success! However, I hope that the author will systematically revise the full writing narrative, which will again improve the quality of the presentation of the manuscript.
AR: We appreciate the reviewer's detailed review and recognition of our work.

Reviewer 2 Report

Dear Authors

I consider the algorithm proposed in your paper to be a significant scientific contribution to Lidar Odometry motion estimation.
There are a few minor errors that should be corrected prior to publication.
That is, misspelled words throughout the text, and in the first image's caption, the subimage (g) is not described.

Author Response

I consider the algorithm proposed in your paper to be a significant scientific contribution to Lidar Odometry motion estimation. There are a few minor errors that should be corrected prior to publication.

AR: We appreciate the reviewer's detailed review and recognition of our work.

Comment 1: That is, misspelled words throughout the text, and in the first image's caption, the subimage (g) is not described.

AR: Thanks for the reviewer's suggestion. We have revised all typos and mistakes in the paper. The changes are marked as blue in the revised manuscript.

Reviewer 3 Report

This paper incorporates intensity information and a dynamic object removal module into the traditional LiDAR odometry frameworks. By doing this, the performance of LiDAR odometry on unstructured scenes is significantly improved. Extensive experiments have been conducted to validate the capability of individual modules and the overall framework.

Major comments:

There are formatting issues in the PDF. Please check it carefully.

Introduction:

Line 55: You can delete the sentence "Besides, we improved ..". This is minor compared to your other contributions.

Figure 1: Good figure, but the problem is that the resolution is too low to really see what features look like in intensity/range/normal images and how well they are labeled and extracted.

Section 3.1.1: How do you get the relative transformation T(e,s)? Maybe you mentioned it in the following section, but it is good to briefly mention it here.

Section 3.1.2 - Scan projection

What is the basic logic behind such projection? Explaining the idea before introducing the details.

You need to mention that 3D coordinates pi are in the laser unit frame.

What are θd, θt, π? How are the width and height of the resulting image defined?

"In D and I, each pixel..." I am not sure what is the meaning of this sentence. Do you mean more than one point will correspond to one pixel? This problem can be solved by changing the width and height of the image, right? Also, what is the logic to choose point with largest reflectance (low reflectance -> noisy points)?

Section 3.1.2 - Object clustering 

Which image is the object clustering performed? S or I or D?

Section 3.1.3 - Feature extraction

Can you give some examples of reflector features in real-world datasets?

What will happen if some features can be classified as two features (e.g., lane marking belongs to ground and also has high intensity, or edge features with high intensity)?

Section 3.1: Overall, the write-up of this subsection is not as clear as those of other subsections. Explaining the ideas/logic first will make it easier to follow.

Figure 5: Can you show sample results of Dynamic points and static points? Also, it will be good to add a box to highlight the captured person.

Figure 8: The color scheme and name of each box are very confusing. You can add some legends explaining the meaning of boxes with a certain color. The reflector features are not clear in this figure (zoom-in images can be shown).

Section 5: The title can be changed to "Conclusions"

Minor comments:

Line 23: abbreviation of LiDAR should come after the full name

Line 26: I think apostrophe marks (') are not required (same for the following text).

Line 70: "a good transfromation initial" -> "a good initial transformation" (same goes for the following, "initial" cannot be used as a noun in such context)

Line 71: "sequentially" -> sequential"

Line 82: Better to have the authors' names before the reference. Same for line 88.

Figure 2: add space between step number and name.

Line 123: "image" -> "images"

Line 159: problem

Line 185: "iamge" -> "image"

Line 286: "efeect" -> "effect"

Author Response

This paper incorporates intensity information and a dynamic object removal module into the traditional LiDAR odometry frameworks. By doing this, the performance of LiDAR odometry on unstructured scenes is significantly improved. Extensive experiments have been conducted to validate the capability of individual modules and the overall framework.   AR: We appreciate the reviewer's detailed review and recognition of our work.

Comment 1: There are formatting issues in the PDF. Please check it carefully.

AR: Thanks for the reviewer's detailed review. We will format the paper in accordance with the requirement of editor.

Comment 2: Line 55: You can delete the sentence "Besides, we improved ..". This is minor compared to your other contributions.

AR: Thanks for the reviewer's suggestion. We have deleted this sentence.

Comment 3: Figure 1: Good figure, but the problem is that the resolution is too low to really see what features look like in intensity/range/normal images and how well they are labeled and extracted.

AR: Thanks for the reviewer's recognition. We have changed pictures in the figure as high-resolution pictures.

Comment 4: Section 3.1.1: How do you get the relative transformation T(e,s)? Maybe you mentioned it in the following section, but it is good to briefly mention it here.

AR: Thanks for the reviewer's suggestion. We have added the description about the term.

Comment 5: Section 3.1.2 - Scan projection: What is the basic logic behind such projection? Explaining the idea before introducing the details.

AR: Thanks for the reviewer's suggestion. We have added the description about the basic logic of projection.   Comment 6: You need to mention that 3D coordinates $\bf{p}_i$ are in the laser unit frame. What are ${\theta}_d$, ${\theta}_t$, $\pi$? How are the width and height of the resulting image defined?
AR: Thanks for the reviewer's suggestion. We have added these description in the paper.

Comment 7: "In D and I, each pixel..." I am not sure what is the meaning of this sentence. Do you mean more than one point will correspond to one pixel? This problem can be solved by changing the width and height of the image, right? Also, what is the logic to choose point with largest reflectance (low reflectance -> noisy points)?  

AR: Thanks for the reviewer's suggestion. You are right, there are more than one point will fall into one specific pixel, thus we need to choose the representative values for the pixel. Here we use the smallest range and the largest reflectance of all inside points, since we focus more on those points near the LiDAR and with high reflectance value. One can also choose other values for their specific application (e.g., the average or mdedian.)

Comment 8: "Section 3.1.2 - Object clustering: Which image is the object clustering performed? S or I or D?
AR: Non-ground range image is used for the object clustering. We added more description in the paper.
Comment 9: "Section 3.1.3 - Feature extraction: Can you give some examples of reflector features in real-world datasets?
AR: Examples of reflector features were displayed in the experimental part in Fig.8. Traffic signs are the most common reflector features in the real-world city scenario.
Comment 10: What will happen if some features can be classified as two features (e.g., lane marking belongs to ground and also has high intensity, or edge features with high intensity)?
AR: During data processing in the code, features are extracted group-by-group, which means that features can only be categorized into one feature group. We extracted ground features first, reflectors second, edge third, and facade last.
Comment 11: Section 3.1: Overall, the write-up of this subsection is not as clear as those of other subsections. Explaining the ideas/logic first will make it easier to follow.   AR: Thanks for the reviewer's suggestion. As mentioned above, we have added the description about the basic logic of projection at the beginning of this part.
Comment 12: Figure 5: Can you show sample results of Dynamic points and static points? Also, it will be good to add a box to highlight the captured person.
AR: Thanks for the reviewer's suggestion. We have circled the captured person in the figure.
Comment 13: Figure 8: The color scheme and name of each box are very confusing. You can add some legends explaining the meaning of boxes with a certain color. The reflector features are not clear in this figure (zoom-in images can be shown).
AR: Thanks for the reviewer's suggestion. We have replaced images in Fig.8 as high-resolution screenshot images.
Comment 14: Minor comments:  Line 23: abbreviation of LiDAR should come after the full name Line 26: I think apostrophe marks (') are not required (same for the following text). Line 70: "a good transformation initial" -> "a good initial transformation" (same goes for the following, "initial" cannot be used as a noun in such context) Line 71: "sequentially" -> sequential" Line 82: Better to have the authors' names before the reference. Same for line 88. Figure 2: add space between step number and name. Line 123: "image" -> "images" Line 159: problem Line 185: "iamge" -> "image" Line 286: "efeect" -> "effect"}} 
AR: Thanks for the reviewer's detailed review. We have checked and revised all these typos and mistakes.